**Observation of new particle formation and measurement of sulfuric acid,**
**ammonia, amines and highly oxidized organic molecules at a rural site in**
**central Germany**
Andreas Kürten[1], Anton Bergen[1], Martin Heinritzi[1], Markus Leiminger[1], Verena Lorenz[1], Felix Piel[1],
Mario Simon[1], Robert Sitals[1], Andrea Wagner[1], and Joachim Curtius[1]
[1]Institute for Atmospheric and Environmental Sciences, Goethe University of Frankfurt, Frankfurt am
Main, Germany
Correspondence to: Andreas Kürten (kuerten@iau.uni-frankfurt.de)
**Abstract**
The exact mechanisms for new particle formation (NPF) under different boundary layer conditions are
not known yet. One important question is if amines and sulfuric acid lead to efficient NPF in the
atmosphere. Furthermore, it is not clear to what extent highly oxidized organic molecules (HOM) are
involved in NPF. We conducted field measurements at a rural site in central Germany in the proximity
of three larger dairy farms to investigate if there is a connection between NPF and the presence of amines
and/or ammonia due to the local emissions from the farms. Comprehensive measurements using a nitrate
Chemical Ionization-Atmospheric Pressure interface-Time Of Flight (CI-APi-TOF) mass spectrometer,
a Proton Transfer Reaction-Mass Spectrometer (PTR-MS), particle counters and Differential Mobility
Analyzers (DMAs) as well as measurements of trace gases and meteorological parameters were
performed. We demonstrate here that the nitrate CI-APi-TOF is suitable for sensitive measurements of
sulfuric acid, amines, a nitrosamine, ammonia, iodic acid and HOM. NPF was found to correlate with
sulfuric acid, while an anti-correlation with RH, amines and ammonia is observed. The anti-correlation
between NPF and amines could be due to the efficient uptake of these compounds by nucleating clusters
and small particles. Much higher HOM dimer (C19/C20 compounds) concentrations during the night
than during the day indicate that these HOM do not efficiently self-nucleate as no night-time NPF is
observed. Observed iodic acid probably originates from an iodine-containing reservoir substance but the
iodine signals are very likely too low to have a significant effect on NPF.
**1. Introduction**

The formation of new particles from gaseous compounds (nucleation) produces a large fraction of atmospheric aerosol particles (Zhang et al., 2012). While the newly formed particles have diameters between 1 and 2 nm they can grow and reach larger sizes, which enables them to act as cloud condensation nuclei (CCN, ~50 nm in diameter or larger). Removal processes such as coagulation scavenging due to larger pre-existing particles can be important if the growth rates ($GR$) for the newly formed particles are slow and/or if the coagulation sink ($CS$) is high. The climatic effect of nucleation depends strongly on the survival probability of the newly formed particles, i.e. if they reach CCN size, or not. Model calculations indicate that nucleation can account for ca. 50% of the CCN population globally (Merikanto et al., 2009). In addition to their climatic effect secondary particles can also influence the human health (Nel, 2005), or reduce visibility, e.g. in megacities (Chang et al., 2009).

New particle formation (NPF) is a global phenomenon and has been observed in many different environments (Kulmala et al., 2004). In most cases a positive correlation with the concentration of gaseous sulfuric acid has been observed (Sihto et al., 2006; Kuang et al., 2008). However, other trace gases, beside $H_2SO_4$ and $H_2O$, need to be involved in the formation of clusters, otherwise the high particle formation rates measured in the boundary layer cannot be explained (Weber et al., 1997; Kirkby et al., 2011). One ternary compound, which enhances the binary nucleation of sulfuric acid and water significantly, is ammonia. However, at the relatively warm temperatures of the boundary layer the presence of ammonia is probably not sufficient for reaching the observed NPF rates when acting together with sulfuric acid and water (Kirkby et al., 2011; Kürten et al., 2016). The same applies for ion-induced nucleation (IIN); the observed IIN rates for the binary and ternary system including ammonia are not high enough to explain the observations (Kirkby et al., 2011). Therefore, recent nucleation experiments focused on organic compounds acting as a ternary compound (beside $H_2SO_4$ and $H_2O$). Many studies indicate that amines have a very strong enhancing effect on nucleation (Kurtén et al., 2008; Chen et al., 2012; Glasoe et al., 2015). Indeed, a chamber experiment could show that the nucleation of sulfuric acid, water and dimethylamine (DMA) at 5°C and 38% RH produced particles at a rate, which is compatible with atmospheric observations in the boundary layer over a relatively wide range of sulfuric acid concentrations (Almeida et al., 2013). For sulfuric acid concentrations $<10^7$ molecule $cm^{-3}$, which are typical for the boundary layer, and dimethylamine mixing ratios of $> \sim 10$ pptv, nucleation was found to proceed at or close to the kinetic limit. This means every collision between sulfuric acid molecules, and clusters associated with DMA, leads to a larger cluster, which does not evaporate significantly (Kürten et al., 2014).

In principle, mass spectrometry using nitrate chemical ionization could be used to detect neutral clusters consisting of sulfuric acid and bases in the atmosphere. However, only few studies indicate that neutral nucleating atmospheric clusters consist of sulfuric acid and ammonia or amines (Zhao et al., 2011; Jiang et al., 2011), while other studies could not identify such clusters (Jokinen et al., 2012; Sarnela et al., 2015). A further outstanding issue is the question about the magnitude of the atmospheric amine mixing ratios at different locations. In the past several years the experimental tools for sensitive

online measurement of amines in the pptv-range became available (Hanson et al., 2011; Yu and Lee
2012). The reported amine levels reach from up to tens of pptv (Hanson et al., 2011; Freshour et al.,
2014; You et al., 2014; Hellén et al., 2014) to < 0.1 pptv (Sipilä et al., 2015). It is therefore an important
question if some of the reported mixing ratios could be biased high or low due to instrumental issues, or
if the natural variability in the amine mixing ratios due to different source strengths can explain the
differences.
Other possible contributors to particle formation are highly oxidized organic compounds (HOM)
originating e.g. from the reaction of monoterpenes with atmospheric oxidants (Zhao et al., 2013; Ehn et
al., 2014; Riccobono et al., 2014; Jokinen et al., 2015; Kirkby et al., 2016; Bianchi et al., 2016). From
this perspective it seems likely that different nucleation pathways are possible and may dominate at
different sites depending, e.g. on the concentration of sulfuric acid, amines, oxidized organic compounds
and other parameters like temperature and relative humidity. Synergistic effects are also possible, e.g. it
has been demonstrated that the combined effect of ammonia and amines can lead to more efficient
particle formation with sulfuric acid and water than for a case where ammonia is not present (Glasoe et
al., 2015). Due to the manifold possibilities for nucleation and the low concentrations of the growing
clusters it is challenging to identify the dominating particle formation pathway from field measurements
in an environment where many possible ingredients for nucleation are present at the same time.
However, such measurements are necessary and previous measurements from Hyytiälä, Finland,
underscored the importance of sulfuric acid, organic compounds and amines regarding NPF (Kulmala
et al., 2013).
In this study, we have chosen to conduct measurements with an emphasis on the observation of NPF
at a rural site in central Germany. The goal of this field campaign was to measure NPF in an amine rich
environment in the vicinity of dairy farms, as cows are known to emit a variety of different amines as
well as ammonia (Schade and Crutzen, 1995; Ge et al., 2011; Sintermann et al., 2014). The
measurements were performed using different particle counters and particle size analyzers as well as
trace gas monitors ($O_3$, $SO_2$ and $NO_x$). A Proton Transfer Reaction-Mass Spectrometer (PTR-MS) is
used to determine the gas-phase concentration of monoterpenes and isoprene, whereas a chemical
ionization time of flight mass spectrometer using nitrate primary ions (Jokinen et al., 2012; Kürten et
al., 2014) is used for the measurement of sulfuric acid, amines, ammonia and highly oxidized organics.
**2. Methods and Measurement Site Description**
**2.1 Measurement Site Description**
The measurement site is located right next to a meteorological weather station operated by the German
Weather Service (DWD measurement station Michelstadt-Vielbrunn/Odenwald, 49°43'04.4" N and
09°05'58.9" E, 452 m a.s.l.). The village Vielbrunn has a total of ~1300 inhabitants and is surrounded
by fields and forests. The next larger cities are Darmstadt (~35 km towards WNW) and Frankfurt/Main
(~50 km towards NNW). The site was chosen for several reasons: (i) three larger dairy farms are close
by, which should possibly enable us to study the effect of amines on new particle formation, (ii) it can
be regarded as typical for a rural or agricultural area in central Europe, (iii) the site is not too far away
from the University of Frankfurt, which allowed to visit the station for instrument maintenance on a
daily basis and (iv) since we could measure right next to a meteorological station infrastructure and
meteorological data from the DWD could be used.
In terms of studying the effect of amines on new particle formation we were expecting to see a direct
effect due to the local emissions from the dairy farms. Each of these farms is keeping a couple of hundred
cows in shelters, which are essentially consisting only of a roof and a fence such that the wind can easily
carry away the emissions. As mentioned in the introduction livestock are known to emit a variety of
amines as well as ammonia (Schade and Crutzen, 1995; Sintermann et al. 2014) both of which should
have an influence on new particle formation and growth (Almeida et al., 2013; Lehtipalo et al., 2016).
The farms are located in the West (~ 450 m distance), South-South-West (~ 1100 m distance) and South-
East (~ 750 m distance) of the station, respectively.
One further aspect that should be considered is the fact that the site is also surrounded by forests
(mixed type of coniferous and deciduous trees, at least 1 km away). Consequently, emissions of, e.g.
monoterpenes ($C_{10}H_{16}$ compounds), can also potentially influence new particle formation as recent
studies indicate that their oxidation products can contribute to NPF and particle growth (Schobesberger
et al., 2013; Riccobono et al., 2014; Ehn et al., 2014; Kirkby et al., 2016).

**2.2 CI-APi-TOF**

The key instrument for the data discussed in this study is the Chemical Ionization-Atmospheric Pressure
interface-Time Of Flight mass spectrometer (CI-APi-TOF). The instrument was first introduced by
Jokinen et al. (2012) and the one used in the present study is described by Kürten et al. (2014). The CI-
APi-TOF draws a sample flow of 8.5 slm (standard liters per minute), which interacts with nitrate
primary ions (($HNO_3)_{0-2}NO_3^-$) within an ion reaction zone at ambient pressure (~ 50 ms reaction time).
The primary ions are generated from the interaction of $HNO_3$ in a sheath gas and a negative corona
discharge (Kürten et al., 2011). The ion source is based on the design by Eisele and Tanner (1993) for
the measurement of sulfuric acid. The primary and product ions are drawn into the first stage of a vacuum
chamber through a pinhole (~350 µm diameter). Quadrupoles in the first and a second stage of the
chamber, operated in rf-only mode, are used to guide the ions. A lens stack in a third stage focuses and
prepares the ions energetically before they enter the time of flight mass spectrometer (Aerodyne
Research Inc., USA and Tofwerk AG, Switzerland). This mass spectrometer has a mass resolving power
of ~4000 Th/Th and a mass accuracy of better than 10 ppm. These characteristics allow the elemental
identification of unknown ions, i.e. different species having the same nominal (integer) $m/z$ ratio can be
separated due to their mass defect. Using isotopic patterns for an expected ion composition supports the
ion identification. For the data analysis the software tofTools (Junninen et al., 2010) is used within the
Matlab environment.
Previous work has shown that the CI-APi-TOF can be used for highly sensitive measurements of
sulfuric acid (Jokinen et al., 2012), clusters of sulfuric acid and dimethylamine (Kürten et al., 2014),
organic compounds with very low volatility (Ehn et al., 2014) and dimethylamine (Simon et al., 2016).
Sulfuric acid and its clusters can be detected after donating a proton to the primary ions, i.e.,
$(H_2SO_4)_n + (HNO_3)_{m=0-2}NO_3^- \rightarrow (H_2SO_4)_{n-1}(HNO_3)_{k=0-2}HSO_4^- + (m-k+1)(HNO_3)$ (R1)
whereas the low volatility organic compounds (HOM) are detected after clustering with $NO_3^-$, i.e.,
$HOM + (HNO_3)_{m=0-2}NO_3^- \rightarrow (HOM)(HNO_3)_{k=0-2}NO_3^- + (m-k)(HNO_3)$. (R2)
In both reactions (R1) and (R2) the presence of water has been neglected for simplicity. The
measurement of amines is possible because they can be associated with nitrate cluster ions (Section 3.6).
Generally, the quantification of a substance is derived with the following equation:

$$concentration = C \cdot ln\left(1 + \frac{\sum product\ ion\ count\ rates}{\sum primary\ ion\ count\ rates}\right).$$ (1)

Equation (1) relates the sum of the product ion count rates to the sum of the primary ion count rates.
Using a calibration constant $C$ the concentration of a neutral substance can be determined. In the case
of the sulfuric acid concentration ($[H_2SO_4]$) the product ion count rates are due to $HSO_4^-$ and
$(HNO_3)HSO_4^-$, while the primary ion count rates include $NO_3^-$, $(HNO_3)NO_3^-$ and $(HNO_3)_2NO_3^-$. The
calibration constant has been determined as $6 \times 10^9$ molecule $cm^{-3}$ (Kürten et al., 2012).
The same calibration constant has also been used for the quantification of HOM. However, in this
case the mass dependent transmission of the CI-APi-TOF was taken into account by the method of
Heinritzi et al. (2016). This requires an additional correction factor in equation (1) which is around 0.4
for the $m/z$ range 300 to 400 Th and 0.22 for the range 500 to 650 Th; these factors take into account
only the transmission as function of the $m/z$ value, while assuming the same ionization efficiency as for
sulfuric acid, which has been shown to be a valid assumption by Ehn et al. (2014). The quantification
of amines will be detailed in Section 3.6. Table 1 gives an overview of the identified ion signals used in
the further analysis evaluating sulfuric acid monomer and dimer concentrations as well as amine,
nitrosamine, ammonia and iodic acid signals (further explanations will be given in the following
sections).
Regarding the loss of sample molecules within the inlet line of the CI-APi-TOF we expect only a
minor effect. As the sample line has a total length around 1 m, a very high flow rate was applied over
most of the inlet length (Berresheim et al., 2000). Only for the last ~15 cm the flow of 8.5 slm was
applied taking the sample from the center part of the first inlet stage where the inlet has a significantly
larger diameter (5 cm instead of 1 cm for the last part) to avoid wall contact of the relevant portion of
the sampled air.

**2.3 PTR-MS**

Volatile Organic Compounds (VOCs) were measured with a calibrated Proton Transfer Reaction-Mass
Spectrometer (PTR-MS using a quadrupole mass spectrometer, IONICON GmbH, Innsbruck, Austria).
The instrument inlet was heated to 60°C and the same temperature was applied to the ion drift tube. The
drift tube was operated at an $E/N$ of 126 Td in order to minimize the formation of protonated water
clusters while maintaining a high sensitivity ($E/N$ is the ratio between the electric field strength $E$ in V
cm$^{-1}$ and the number density $N$ of gas molecules in cm$^{-3}$, see Blake et al., 2009).
A calibration of the instrument was performed prior to the campaign with a gas mixture containing
several VOCs at a known volume mixing ratio (Ionimed-VOC-Standard, Innsbruck, Austria), including
isoprene, α-pinene, and acetone amongst others. The calibration was performed for a relative humidity
range of 0 to 100 % (steps of 20 %) at room temperature. However, especially for α-pinene (measured
at 81 and 137 Th), the sensitivity of the PTR-MS operating at the rather high $E/N$ was not depending on
relative humidity. For isoprene (measured at 41 and 69 Th), a higher RH led to lower fragmentation
inside the instrument, but this did not affect the overall sensitivity much (<5 % decrease from 20 to

202 100%).

The PTR-MS cannot readily distinguish between different monoterpenes as all have the same
molecular weight, so only the sum of monoterpenes could be measured. However, since α-pinene is
often the most abundant monoterpene in continental mid latitudes (Geron et al., 2000; Janson and de
Serves, 2001) and the reaction rate constants for different monoterpenes are rather similar (Tani et al.,
2003; Cappellin et al., 2012) our estimation of total monoterpene concentration should not be affected
by large errors.

**2.4 Other instrumentation**

Trace gas monitors were used to measure the mixing ratios of sulfur dioxide (Model 43i TLE Trace
Level $SO_2$ Analyzer, Thermo Scientific), ozone (Model 400, Ozone Monitor, Teledyne API) and
nitrogen oxides ($NO_x$, Ambient $NO_x$-Monitor APNA-360, Horiba). These instruments were calibrated
once before the campaign with known amounts of trace gases and dry zero air was applied on a daily
basis for a duration of at least half an hour in order to take instrument drifts into account. The detection
limits of the gas monitors are 0.05 ppbv for the $SO_2$ monitor (for a 5 minute integration time),
approximately 0.5 ppbv for the $NO_x$ monitor and 0.5 to 1 ppbv for the $O_3$ monitor.
Further instruments used include condensation particle counters (CPCs) and differential mobility
analyzers (DMAs). The CPCs 3025A and 3010 (TSI, Inc.) were used to determine the total particle

concentration above their cut-off sizes of 2.5 and 10 nm, respectively. A Scanning Mobility Particle Sizer (SMPS) from TSI (Model 3081 long DMA with a CPC 3776) determined the particle size distribution between 16 and 600 nm. The smaller size range was covered by a nDMA (Grimm Aerosol Technik, Germany) and a TSI CPC 3776 for diameters between 3 and 40 nm. The combined size distribution can be used to calculate the condensation/coagulation sink towards certain trace gases (e.g. sulfuric acid) or particle diameters.

Meteorological parameters were both obtained from our own measurements with a Vaisala sonde (Model WXT 520), which yielded the temperature, RH, wind speed and direction as well as the amount of precipitation. The same parameters are also available for the Vielbrunn meteorological station from the DWD; additionally, values for the global radiation were provided from the DWD.

**3. Results**

**3.1 Meteorological conditions and overview**

The intensive phase of the campaign was from May 18 to June 7, 2014 (21 campaign days). Figure 1 shows an overview of the meteorological conditions, i.e. temperature, relative humidity, global radiation and precipitation. The size distribution of small particles (Fig. 1, bottom panel) was measured by the nDMA. In addition, the condensation sink calculated for the loss of sulfuric acid on aerosol particles is also shown taking into account the full size distribution (up to 600 nm).

The first part of the campaign (including May 22) was characterized by warm temperatures and sunny weather without precipitation. Between May 22/23 and May 31 the weather conditions were less stable with colder temperatures and some precipitation events. Especially on May 29 a strong drop in temperature and the condensation sink was observed, due to a cold front followed by the passage of relatively clean air. From May 31 on temperatures were increasing again and it was mostly sunny with only two rain events on June 3 and June 4.

Elevated concentrations of small particles could be observed on almost every day. However, new particle formation from the smallest sizes (around 3 nm) followed by clear growth were seen only on 6 days out of 21 (i.e. 29%). These events, which were also used for the calculation of new particle formation rates (Section 3.9), are highlighted in the bottom panel of Fig. 1 by the dark gray arrows. The presence of small particles was also observed on several other days, however, the events were either relatively weak, or no clear particle growth was observable.

**3.2 Trace gas measurements**

The trace gas measurements are shown in Fig. 2. Typical maximum day-time ozone mixing ratios ranged
from ~40 to 75 ppbv (Fig. 2, upper panel). The sulfur dioxide levels were between 0.05 and a maximum
of 2 ppbv with average values around 0.3 ppbv (Fig. 2, upper panel). Especially during the passage of
clean air on May 29 and May 30 the $SO_2$ levels were quite low. $NO_2$ mixing ratios showed a distinct
diurnal pattern with a minimum in the late afternoon and an average mixing ratio around 3 ppbv (Fig.
2, middle panel, see also Fig. 8). The NO mixing ratios were about a factor of 5 lower compared to $NO_2$
on average (Fig. 2, middle panel, see also Fig. 8); similar values were reported for another rural site in
Germany (Mutzel et al., 2015). The maximum sulfuric acid concentrations were reached around noon
and ranged between $\sim 1 \times 10^6$ and $2 \times 10^7$ molecule cm$^{-3}$ (Fig. 2, lower panel, see also Fig. 3), which is
comparable to other sites (Fiedler et al., 2005; Petäjä et al., 2009). The total monoterpene and isoprene
mixing ratios measured by the PTR-MS were similar to each other with values between ~0.03 and 1
ppbv (Fig. 2, lower panel). Mixing ratios in the same range have also been reported for the boreal forest
(Rantala et al., 2014).

**3.3 $H_2SO_4$ measurement and calculation from proxies**

Figure 3 shows the average diurnal sulfuric acid concentration along with other data, which will be
discussed in later sections. The maximum average $[H_2SO_4]$ around noon was $\sim 3 \times 10^6$ molecule cm$^{-3}$; the
error bars represent one standard deviation.
Recently, Mikkonen et al. (2011) introduced approximations to calculate sulfuric acid as a function
of different proxies. Since the relevant parameters (sulfur dioxide mixing ratio, global radiation, relative
humidity and condensation sink) are available, we have used the following formula to approximate the
sulfuric acid concentration (Mikkonen et al., 2011):

$$[H_2SO_4]_{proxy} = a \cdot k(T,p) \cdot [SO_2]^b \cdot Rad^c \cdot RH^d \cdot CS^e. \tag{2}$$

The $[H_2SO_4]$ (expressed in molecule cm$^{-3}$) is calculated as a function of the $SO_2$ mixing ratio (in ppbv),
the global radiation $Rad$ (in W m$^{-2}$), the relative humidity $RH$ (in %), the condensation sink $CS$ (in s$^{-1}$),
a rate constant $k$, which depends on ambient pressure $p$ and temperature $T$ (see definition for $k$ by
Mikkonen et al., 2011) and a scaling factor $a$. A least square fit made with the software IGOR yields the
coefficients $a = 1.321 \times 10^{15}$, $b = 0.913$, $c = 0.990$, $d = -0.217$ and $e = -0.526$ (linear correlation coefficient,
Pearson's $r$, is 0.87). Following the recommendations given by Mikkonen et al. (2011) we restricted the
data used in the derivation of the parameters to conditions where the global radiation was equal or larger
than 50 W m$^{-2}$. In addition, a simpler formulation was also tested, which neglects the dependence on $RH$
and $CS$:

$$[H_2SO_4]_{proxy'} = a' \cdot k(T,p) \cdot [SO_2]^{b'} \cdot Rad^{c'}. \tag{3}$$

Here, the coefficients $a' = 1.343 \times 10^{16}$, $b' = 0.786$ and $c' = 0.941$ yield good agreement (linear correlation
coefficient, Pearson's $r$, is 0.85) between calculated and measured [$H_2SO_4$]. Figure 4 shows a
comparison between the two approximation methods and the measured sulfuric acid for the full
campaign (when $Rad \geq 50$ W m$^{-2}$). In almost all cases the predicted 5 minute averages are within a factor
of 3 of the measured values for both methods. This indicates that even the simpler method (equation (3))
can yield relatively accurate results for the conditions of this study. This is probably due to the fact that
$RH$ and $CS$ show only relatively small variations over the duration of the campaign and it would
therefore not be absolutely necessary to include these factors in the sulfuric acid calculation.
Nevertheless, whenever the data are available we recommend to use the more detailed parameterization
(equation (2)) as it treats the sulfuric acid concentration calculation more rigorously. The parameters
found are in good agreement with the ones reported by Mikkonen et al. (2011) for different sites.

**3.4 Calculated OH**

For further data evaluation knowledge of the OH concentrations is useful. In this study the hydroxyl
radical concentration is required to derive an estimated concentration of the iodine species OIO (Section
3.5) and for a comparison of conditions during nucleation and no nucleation days (Section 4). Since
there was no direct measurement of the hydroxyl radical available, only an estimation based on other
measured parameters can be made. This estimation is based on the assumption that most of the sulfuric
acid is produced from the reaction between $SO_2$ and OH. Using the condensation sink $CS$ the balance
equation between production and loss at steady-state can be used to derive the OH:

$$[\text{OH}] \quad = \frac{CS \cdot [\text{H}_2\text{SO}_4] - k_{\text{X}+\text{SO}_2} \cdot [\text{X}] \cdot [\text{SO}_2]}{k_{\text{OH}+\text{SO}_2} \cdot [\text{SO}_2]} \approx \frac{CS \cdot [\text{H}_2\text{SO}_4]}{k_{\text{OH}+\text{SO}_2} \cdot [\text{SO}_2]}. \tag{4}$$

Recently it was discovered that there are also other species capable of oxidizing $SO_2$ to $SO_3$ (which lead
to subsequent production of $H_2SO_4$ due to further reactions with $O_2$ and $H_2O$) (Mauldin et al., 2012).
Those species X, e.g. stabilized Criegee Intermediates (sCI) can be formed via the ozonolysis of alkenes
(e.g. isoprene, α-pinene, limonene) (Mauldin et al., 2012; Berndt et al., 2014). Therefore, if some $H_2SO_4$
is generated from sCI reactions with $SO_2$, then the calculated OH is an upper estimate. During the day
this effect should be relatively small, i.e. < 50% (Boy et al., 2013; Sarwar et al., 2013), although Berndt
et al. (2014) state that no final answer can be given regarding the effect of the sCI on the sulfuric acid
formation because it depends strongly on the sCI structure and competitive reactions between sCI and
water vapor. However, it should also be noted that the reaction between alkenes and ozone generates
not only sCI but also OH at significant yields (e.g. the OH yield for the reaction of α-pinene and $O_3$ is
ca. 0.77, Forester and Wells, 2011) and that the OH produced via this mechanism is taken into account
by equation (4). Data from Sipilä et al. (2014) further suggest that the production of sulfuric acid from
sCI oxidation of $SO_2$ is probably minor compared to that from OH and $SO_2$. For these reasons we have
decided to calculate the [OH] not only for the day time but for the full day. The derived diurnal pattern
of [OH] is shown in Fig. 3 with a maximum concentration of $1\times10^6$ molecule $cm^{-3}$ around noon, which
is in good agreement with other studies where OH was measured directly (Berresheim et al., 2000;
Rohrer and Berresheim, 2006; Petäjä et al., 2009).

**3.5 Iodic acid ($HIO_3$) and OIO**

The high resolution CI-APi-TOF mass spectra revealed the presence of iodine containing substances. It
can be ruled out that these signals result from instrument contamination as our CI-APi-TOF had never
been in contact with iodine (i.e. no nucleation experiments with iodine have yet been performed and no
iodide primary ions have been used). The observed signals could be assigned to $IO_3^-$, $(H_2O)IO_3^-$ and
$(HNO_3)IO_3^-$ (Table 1). To our knowledge the identification of iodine related peaks have not been
reported from measurements with a nitrate CIMS. However, Berresheim et al. (2000) reported the
presence of a peak at $m/z$ 175 in the spectrum for the marine environment, which was not identified
previously but in the light of this study, can almost certainly be attributed to $IO_3^-$.

347        The diurnal pattern of $IO_3^-$ and the related iodine peaks show a distinct pattern with a maximum

around noon following almost perfectly the diurnal pattern of sulfuric acid (Fig. 3). This may not be
surprising since the formation of $HIO_3$ is due to reaction between OIO and OH (Saiz-Lopez et al., 2012);
therefore, the iodic acid concentration is connected to the OH chemistry. After normalization of the
iodic acid signals with the nitrate primary ion count rates, a concentration of the neutral compound $HIO_3$
can be obtained by tentatively adopting the same calibration constant for iodic acid as for sulfuric acid.
Thereby a maximum average day-time concentration of ~$3\times10^5$ molecule $cm^{-3}$ can be found. Further
using the derived OH concentrations from the $H_2SO_4$ and $CS$ measurements (Section 3.4) the derived
[$HIO_3$] can be used to estimate the concentration of OIO (Saiz-Lopez et al, 2012):

$$[\text{OIO}] = \frac{cs \cdot [\text{HIO}_3]}{k_{\text{OH+OIO}} \cdot [\text{OH}]}.\tag{5}$$

Equation (5) assumes that the only production channel of $HIO_3$ is the reaction between OH and OIO and
the only loss mechanism of $HIO_3$ is the uptake on aerosol. The reaction rate $k_{\text{OH+OIO}}$ can be taken from
the literature (Plane et al., 2006). In this way the concentration of OIO can be estimated to a typical
value of $5\times10^6$ molecule $cm^{-3}$, which is much lower than the values reported for the marine environment
(3 to 27 pptv, i.e. $7.5\times10^7$ to $6.8\times10^8$ molecule $cm^{-3}$, see Saiz-Lopez et al., 2012).

364        The relatively low values of [$HIO_3$] and [OIO] probably indicate that iodine chemistry is not very

important in terms of new particle formation at this site. This is supported by the fact that we could not
observe any clusters containing e.g. sulfuric acid and iodic acid or clusters containing more than one
iodine molecule. However, it is surprising that iodine can be detected more than 400 km away from the
nearest coast line. On the other hand, HYSPLIT back trajectory calculations (Stein et al., 2015) reveal
that in most cases the air was arriving from westerly directions and therefore had contact with the ocean
within the last 48 hours before arriving at the station. During the measurement period there was
unfortunately never a day where the air was clearly coming from easterly directions and had not been in
contact with the Atlantic Ocean or Mediterranean Sea within the previous days. Therefore, it could not
be checked if this would result in lower iodine signals. Despite the marine origin of the air masses
observed it is not clear how the iodine is transported over relatively large distances without being lost
on aerosol particles. If iodic acid is irreversibly lost on aerosol (similar to sulfuric acid) its lifetime
should only be on the order of several minutes at typical boundary layer conditions. Therefore, the
presence of iodine indicates either a local iodine source, or its transport from marine environments in
the form of a reservoir substance, e.g. $CH_3I$ (the lifetime of $CH_3I$ is in the order of 1 week, see Saiz-
Lopez et al., 2015), and subsequent release due to photolysis.
Regarding the sensitivity of the CI-APi-TOF it can be said that iodic acid (and, if present, probably
also its clusters) can be detected with high sensitivity. One aspect that helps in unambiguously
identifying iodic acid is the high negative mass defect of the iodine atom ($\Delta m \approx$ -0.1 Th). Furthermore,
this also contributes to the low detection limit for this compound because generally there will not be any
overlapping signals from other substances having the same integer mass (mass resolving power of the
instrument is ~4000 Th/Th, i.e. at $m/z$ 175 the peak width at half maximum is ~0.04 Th). The method
introduced here therefore allows high-sensitivity measurement of $[HIO_3]$ as well as the estimation of
$[OIO]$ with the help of equation (5) in future studies. The lowest detectable concentrations should be
around $3 \times 10^4$ molecule $cm^{-3}$, or better, for $[HIO_3]$ and $5 \times 10^5$ molecule $cm^{-3}$ (ca. 0.02 pptv) for $[OIO]$
when assuming the same calibration constant for $HIO_3$ as for $H_2SO_4$ and considering the lowest iodine
signal from Fig. 3.
**3.6 Amine, nitrosamine and ammonia measurements**
The detection of dimethylamine (DMA, $(CH_3)_2NH$) by means of nitrate chemical ionization with a CI-
APi-TOF has been described previously (Simon et al., 2016). The clustering between diethylamine
(DEA) and nitrate ion clusters has also been reported by Luts et al. (2011). The amines detected in the
present study include $CH_5N$ (monomethylamine), $C_2H_7N$ (dimethylamine, DMA or ethylamine, EA),
$C_3H_9N$ (trimethylamine, TMA or propylamine, PA), $C_4H_{11}N$ (diethylamine, DEA) and $C_6H_{15}N$
(triethylamine, TEA). All these amines are identified as clusters in the CI-APi-TOF spectra where the
amines are associated both with the nitrate dimer ($(amine)(HNO_3)NO_3^-$) and the trimer
($(amine)(HNO_3)_2NO_3^-$).
The high mass resolving power of the CI-APi-TOF allowed the identification of five different amines
(C1-, C2-, C3-, C4- and C6-amines, see above). Since the amines are all identified at two different
masses each (either with the nitrate dimer or the nitrate trimer) plotting the time series of each pair of
signals allows further verification of the amine signals since a different time trend would reveal that
another ion would interfer with the amine signal. This was sometimes the case when the relative
humidity was high and clusters of water and nitrate appeared with high water numbers. The cluster of
$NO_3^-$ and 6 water molecules has a mass of 170.0518 Th and the C2-amine cluster $(C_2H_7N)(HNO_3)NO_3^-$
(170.0419 Th) cannot be separated from this primary ion cluster. Therefore, if large nitrate plus water
clusters were observed in the spectra, no C2-amine signal could be evaluated.
The same ion cluster chemistry applies for ammonia, which can also bind with the nitrate cluster
ions. Consequently, ammonia is detected as $(NH_3)(HNO_3)NO_3^-$ and $(NH_3)(HNO_3)_2NO_3^-$ (Table 1). To
our knowledge the existence of these cluster ions has not been reported previously.
In accordance with Simon et al. (2016) the cluster ion signals have been normalized by the following
relationship:
$$\text{amine}_{ncps} = \ln\left(1 + \frac{\{(\text{amine})(HNO_3)NO_3^-\} + \{(\text{amine})(HNO_3)_2NO_3^-\}}{\{(HNO_3)_2NO_3^-\}}\right), \tag{6}$$
where the curly brackets denote the count rates of the different ion clusters and the same formula can be
used when "amine" is replaced by $NH_3$ to obtain the normalized ammonia signal. The normalization
with the nitrate trimer has been chosen because we think that this is the dominant nitrate ion cluster the
amines (and ammonia) can bind to within the CI-APi-TOF ion reaction zone (Simon et al., 2016). Partial
evaporation of one $HNO_3$ from the resulting amine nitrate cluster within the CI-APi-TOF vacuum
chamber leads to the spread of the signal over the related masses separated by 62.9956 Th ($HNO_3$).
In addition, to the five amines mentioned before, we were able to identify dimethylnitrosamine
(NDMA, $(CH_3)_2NNO$) from its clusters $((CH_3)_2NNO)(HNO_3)NO_3^-$ and $((CH_3)_2NNO)(HNO_3)_2NO_3^-$
(Table 1). The signals from NDMA show a clear diurnal pattern on some days, which can be up to about
two orders of magnitude higher during the night compared to the day. This is in agreement with the
formation mechanism of NDMA via the reaction of DMA with OH and NO (Nielsen, Herrmann and
Weller, 2012). The lower concentrations during the day can be explained by the rapid photolysis rate of
NDMA (Nielsen, Herrmann and Weller, 2012). Since only C2-amines are capable of forming
nitrosamines no further nitrosamine could be identified from the mass spectra. Only a rough estimation
of the mixing ratio can be provided by using the calibration constant from Simon et al. (2016) which
was derived for DMA. Using this calibration constant the maximum mixing ratio of NDMA would be
~100 pptv (or $2.5 \times 10^9$ molecule cm$^{-3}$). However, this value has a high uncertainty because no direct
calibration with NDMA was performed.
The average diurnal patterns of the four amines and ammonia are shown in Fig. 5. The data are an
average over 21 measurement days and the error bars represent one standard deviation. The temperature
profile is shown along with the CI-APi-TOF signals. The C4-, C6-amines and ammonia show a distinct
diurnal profile, which follows the temperature profile closely. The temperature-dependent signal
intensity could be due to partial re-evaporation of amines from the particulate phase. No correlation with
temperature is seen for the C1-, C2- and C3-amines, which could indicate efficient stabilization of these
amines in the particulate phase due to acid-base reactions (Kirkby et al., 2011; Almeida et al., 2013).
No direct calibration for amines, NDMA and ammonia was performed during the campaign.
Therefore, only a rough estimation of the mixing ratios can be made. Using the calibration curve for
DMA by Simon et al. (2016), $1\times10^{-4}$ ncps (normalized counts per second) correspond to ~1 pptv of
DMA. With this conversion the average mixing ratios are between about 1 and 5 pptv for the amines.
The mixing ratios from this study are in a similar range as those reported from measurements in a
southeastern US forest (You et al., 2014) but generally lower as those from three different sites in the
US (Freshour et al., 2014).
The ncps for ammonia are lower than for the amines, which should not be the case if the sensitivity
towards ammonia and amines would be the same because the ammonia mixing ratios are almost
certainly higher than the ones for the amines in this environment. The ammonia mixing ratio can be
above several ppbv in rural areas (von Bobrutzki et al., 2010). Therefore, the sensitivity of the nitrate
CI-APi-TOF towards ammonia seems to be significantly lower than for amines. This is reasonable, since
other studies found that acid-base clusters between sulfuric acid (including the bisulfate ion) and amines
are much more stable compared to sulfuric acid ammonia clusters (Kirkby et al., 2011; Almeida et al.,
2013). Therefore, the acid base clustering between nitric acid (including the nitrate ion) and ammonia
or amines could follow a similar rule, which would lead to faster evaporation of the ammonia nitrate
clusters. For this reason only the relative signals for ammonia can be used at the moment without
providing estimated mixing ratios.
Recently it has been suggested that diamines could play an important role in ambient NPF (Jen et al.,
2016a); however, we could not identify diamines from the high-resolution mass spectra.

**3.7 Sulfuric acid dimer**

Occasionally, the CI-APi-TOF sulfuric acid dimer signal was above background levels. The dimer
$((H_2SO_4)HSO_4^-)$ was identified from the high resolution spectra on nine campaign days. The measured
sulfuric acid dimer concentrations are shown as a function of the sulfuric acid monomer concentrations
in Fig. 6. For comparison, CLOUD chamber data from nucleation experiments in the ternary sulfuric
acid-water-dimethylamine system are included (red circles in Fig. 6, Kürten et al., 2014). In addition,
the lower dashed line shows the expected dimer formation due to ion-induced clustering (IIC) of sulfuric
acid monomers in the CI-APi-TOF ion reaction zone (Hanson and Eisele, 2002; Zhao et al., 2010).
The data indicate that the measured dimer concentrations are clearly above the background level set
by ion-induced clustering. On the other hand the concentrations are lower than what has been measured
in CLOUD for kinetic nucleation in the sulfuric acid-water-dimethylamine system at 5°C and 38% RH
(Almeida et al., 2013; Kürten et al., 2014). Clearly, the neutral sulfuric acid dimers were stabilized by a
ternary compound, otherwise their concentrations would not have been measurable at these relatively
warm conditions because the dimer (without a ternary compound) evaporation rate is quite high ($> 10^5$
s$^{-1}$ at 290 K, Hanson and Lovejoy, 2006; Kürten et al., 2015). On the other hand the ternary stabilizing
agent evaporates after charging of the sulfuric acid dimers because no cluster between the sulfuric acid
dimer and another compound (besides $HNO_3$ from the ion source) could be identified. This means that
although the dimers contained at least one additional molecule in the neutral state, the ionized dimer
will be detected as $(H_2SO_4)HSO_4^-$ (Ortega et al., 2014; Jen et al., 2014), which makes it impossible to
identify the stabilizing agent. Only when larger clusters of sulfuric acid are present (trimer and larger)
stabilizing agents like ammonia or amines can stay in the cluster after charging with the nitrate ion (Zhao
et al., 2011; Kirkby et al., 2011; Ortega et al., 2014; Kürten et al., 2014). Unfortunately, no large sulfuric
acid clusters (trimer and larger) were measurable during the campaign, probably because their
concentrations were too low. Therefore, only speculations about the stabilizing agent responsible for the
high dimer concentrations can be made. It is quite unlikely that ammonia would be the only stabilizing
compound for the dimers since previous studies have shown that the relatively high dimer concentration
measured at rather low sulfuric acid monomer concentrations ($< 2\times10^7$ molecule cm$^{-3}$) cannot be
explained by sulfuric acid-ammonia-water nucleation (Hanson and Eisele, 2002; Jen et al., 2014). In
addition, efficient clustering between sulfuric acid and iodic acid can probably be ruled out (provided
that these compounds would be capable of producing a cluster with a low evaporation rate) as the
concentrations of iodic acid are quite low (~ $3\times10^5$ molecule cm$^{-3}$ at maximum, see Section 3.5). This
means that the arrival rate of iodic acid on a sulfuric acid dimer is on the order of $10^{-4}$ s$^{-1}$ (using a
collision rate of $5\times10^{-10}$ cm$^3$ molecule$^{-1}$ s$^{-1}$). Due to the high evaporation rate of the pure sulfuric acid
dimer no significant dimer stabilization by iodic acid can be expected.
Whether amines are responsible for the dimer formation in the present study cannot be concluded. If
they were, the lower dimer concentrations compared to the CLOUD chamber results (Kürten et al.,
2014) could be attributed to the higher temperatures in the present study, which result in faster
evaporation rates. Another explanation would be the lower amine mixing ratios. In the CLOUD study
dimethylamine was present at 10 pptv, or higher. In addition, it cannot be concluded that e.g. the
measured C2-amines are all dimethylamine, if a significant fraction of them were, e.g., ethylamine, its
stabilizing effect could be significantly lower. This remains somewhat speculative as no data regarding
NPF from ethylamine and sulfuric acid was found, however, triethylamine was reported to have a
relatively weak effect on nucleation compared to DMA or TMA (Glasoe et al., 2015). Other compounds
which are present and have been shown to form new particles are HOM (Schobesberger et al., 2013;
Ehn et al., 2014; Riccobono et al., 2014;) although their stabilizing effect on neutral sulfuric acid dimers
remains to be elucidated.
Regarding the observations shown in Fig. 6, it should be noted that no ion filter (high voltage electric
field in the CI-APi-TOF inlet to remove ambient ions) was used in the present study. This could in
principle lead to the detection of ambient ions and clusters, which did not undergo charging in the CI-
APi-TOF ion reaction zone. If this were the case, no representative concentrations of the corresponding
neutral sulfuric acid dimer would be derived. CLOUD studies reported that charged sulfuric acid
monomers ($HSO_4^-$) and dimers (($H_2SO_4$)$HSO_4^-$) could be observed with a different nitrate chemical
ionization mass spectrometer (CIMS) under some conditions (Rondo et al., 2014; Kürten et al., 2015).
However, for ambient measurements, no significant effect could be observed for sulfuric acid monomers
(Rondo et al., 2014). In principle, the sulfuric acid dimer could be more strongly affected by the
detection of ambient ions since the neutral dimer concentration is much lower than the sulfuric acid
monomer, while the negative ambient ion spectrum can be dominated by the charged sulfuric acid dimer
(Eisele et al., 2006). Therefore, we cannot entirely rule out that ambient ions had some effect on the data
shown in Fig. 6. However, the ambient ions would need to overcome an electric field before they could
enter the ion reaction zone (Kürten et al., 2011; Rondo et al., 2014). In the CIMS and the CI-APi-TOF
a negative voltage is used to focus the primary ions to the center of the reaction zone, while the sample
line is electrically grounded. This means negative ambient ions would need to overcome a repulsing
electric field which acts as a barrier. Light ions will be efficiently deflected due to their high mobility
but heavier ions can in principle penetrate more easily. Consequently, CIMS measurements at the
CLOUD chamber showed that the apparent dimer signal measured by the CIMS correlated with large
ion clusters (pentamer, i.e. ($H_2SO_4$)$_4HSO_4^-$ and larger, which underwent subsequent fragmentation) but
not with the ($H_2SO_4$)$HSO_4^-$ signal; the charged clusters were measured simultaneously with an APi-
TOF (Junninen et al., 2010; Kürten et al., 2015). The CI-APi-TOF used in this study utilized a higher
voltage for the ion focusing compared to the CIMS (ca. -500 V instead of -220 V in the CIMS) and
should therefore prevent smaller masses even more efficiently from entering the ion source than in the
study by Kürten et al. (2015). In addition, the absence of any trimer signal (($H_2SO_4$)$_2HSO_4^-$) in the
spectra argues against ambient ion detection. In a previous study by Eisele et al. (2006) ambient ion
measurements showed, besides signals for ($H_2SO_4$)$HSO_4^-$, also signals for ($H_2SO_4$)$_2HSO_4^-$ which were
on average ~50% of the dimer signals. Since the CI-APi-TOF design, with its repulsing voltages towards
ambient ions in the ion reaction zone, should be more sensitive towards the trimer than towards the
dimer, the absence of sulfuric acid trimer signals argues against a significant bias in the data due to
charged ambient clusters.

**3.8 Highly oxidized organic molecules (HOM)**

Recently, the rapid autoxidation of atmospherically relevant organic molecules, such as isoprene and
monoterpenes, was described (Crounse et al., 2013; Ehn et al., 2014). There is evidence that these HOM
are involved in the formation of secondary aerosol and can even promote the formation of new aerosol
particles (Jokinen et al., 2015; Kirkby et al., 2016). Nitrate chemical ionization mass spectrometry is
capable of detecting a suite of HOM when the O:C-ratio is high (e.g. > ~0.6 for C10 and > ~0.35 for
C19/C20 compounds) through association of an $NO_3^-$ primary ion (Ehn et al., 2014), while other
ionization techniques are more selective towards less oxidized compounds (Aljawhary et al., 2013).
Many recent publications report peak lists for different compounds identified from chamber or ambient
measurements with nitrate chemical ionization (Ehn et al., 2012; Kulmala et al., 2013; Ehn et al., 2014;
Mutzel et al., 2015; Praplan et al., 2015; Jokinen et al., 2015; Kirkby et al., 2016). The species from the
previous studies are mainly C10 (containing 10 carbon atoms) or C20 (containing 19 or 20 carbon atoms)
compounds originating from reactions between monoterpenes (in most cases from α-pinene) and ozone
and/or OH.

559        The C10 compounds can be further segregated in HOM radicals ($RO_2$, i.e. $C_{10}H_{15}O_{i\geq6}$), HOM

monomers ($C_{10}H_{14}O_{i\geq7}$ and $C_{10}H_{16}O_{i\geq6}$) and HOM involving reactions with nitrate ($C_{10}H_{15}NO_{i\geq7}$ and
$C_{10}H_{16}N_2O_{i\geq8}$) (Jokinen et al., 2014). Dimers (C19/C20 compounds) originate from reactions among
HOM $RO_2$ radicals (Ehn et al., 2014).

563        The spectra were evaluated according to the peak list shown in Table 2 regarding HOM. It should be

noted that the listed compounds represent some fraction of the observed signal in the monomer and
dimer region although not all of the peaks that are present are identified yet. Figure 7 shows a comparison
between the average day time and the average night time spectra for the mass to charge range between
$m/z$ 300 and 650. According to Fig. 7 the main difference between day and night are the significantly
higher signals in the dimer region during the night.

569        Fig. 8 shows the diurnal variation of the HOM (separated into HOM radicals, HOM monomers, HOM

nitrates and HOM dimers according to Table 2) together with other parameters (NO, $NO_2$, $O_3$ and global
radiation). One striking feature is the pronounced maximum concentration of HOM dimers during the
night. During the day when the global radiation shows values above zero the dimer signals drop by about
one order of magnitude and reach levels, which are close to the detection limit of the instrument. The
low day-time dimer concentrations are probably due to enhanced NO, $HO_2$ and $R'O_2$ concentrations
during the day. These compounds can react with HOM $RO_2$ radicals and thereby inhibit the formation
of dimers; which are a result of the reaction between two $RO_2$ radicals. As can be seen from Fig. 8 the
NO concentration peaks in the morning. $HO_2$ was not measured but typically peaks around noon or in
the later afternoon (Monks, 2005). Direct photolysis of HOM dimers has to our knowledge not been
reported in the literature but could in principle also explain the dimer pattern seen in Fig. 8.

580        The HOM monomer signal (Fig. 8) does not show a pronounced diurnal cycle, only in the early

morning the signals are reduced by about 50% compared to the daily average. Slightly higher values
around noon could be explained by the higher $O_3$ and OH concentrations during mid-day, which lead to
enhanced formation of HOM through reactions between these compounds and monoterpenes (Jokinen
et al., 2015; Kirkby et al., 2016). The HOM nitrates, di-nitrates and radicals show almost the same
profile as the HOM monomers. This might be expected for the HOM radicals as these can be regarded

as the precursors for the HOM monomers but the fact that the HOM nitrates follow an almost identical pattern is somewhat surprising as the NO mixing ratio shows a different profile and is thought to be involved in the formation of the HOM nitrates. However, further involvement of e.g. OH, $HO_2$ and $R'O_2$ in the HOM formation should also play a role and therefore influence their diurnal pattern. The elucidation of the HOM formation mechanisms is beyond the scope of this article and will therefore not be discussed further. More field and chamber experiments are needed to identify the influence of different trace gases and radicals on the formation and concentration of HOM.

**3.9 Particle formation rates**

The presence of small particles (< ~20 nm) was observed on almost every day during the campaign. However, often nanometer-sized particles appeared suddenly without clear growth from the smallest size the nDMA covered (slightly above 3 nm). In total there were seven events where clear growth was detectable and these events were the only ones for which a new particle formation rate ($J$) was derived. It should be noted that clear NPF was observed only on 6 days, however, for one day two NPF rates were derived, which results in a total of 7 NPF rates.

In accordance with other previous studies (Metzger et al., 2010; Kirkby et al., 2011) we have first derived a new particle formation rate at a larger mobility diameter $d_{p2}$ (2.5 nm in the present study), which was corrected to a smaller diameter of $d_{p1} = 1.7$ nm in a second step. The formation rate $J_{dp2}$ is obtained from the time derivative of the small particle concentration, which follows from the difference in particle concentrations ($N_{2.5-10}$) measured by the TSI 3776 (cut-off diameter of 2.5 nm) and a TSI 3010 (cut-off diameter of 10 nm):

$$J_{d_{p2}} = \frac{dN_{2.5-10}}{dt} + CS_{d_{p2}} \cdot N_{2.5-10} + \frac{GR}{10nm-2.5nm} \cdot N_{2.5-10}. \tag{7}$$

The second term on the right-hand side in equation (7) accounts for the loss of small particles on particles larger than 2.5 nm, while the third term accounts for the growth of particles out of the size range under consideration (Kulmala et al., 2012). The coagulation sink $CS_{dp2}$ is calculated from the particle size distribution measured by the nDMA and the SMPS. The second step involves an exponential correction to obtain the particle formation rate at the smaller size, $J_{dp1}$, by taking into account the coagulation sink and the growth rate ($GR$) of particles (Lehtinen et al., 2007):

$$J_{d_{p1}} = J_{d_{p2}} \cdot exp\left(\frac{CS_{d_{p1}}}{GR} \cdot dp_1 \cdot \gamma\right). \tag{8}$$

The factor $\gamma$ is defined as follows (Lehtinen et al., 2007):

$$\gamma = \frac{1}{s+1} \cdot \left( \left( \frac{d_{p2}}{d_{p1}} \right)^{s+1} - 1 \right),$$ (9)

where $s$ is the slope of the coagulation sink as a function of size for the size range between $d_{p1}$ and $d_{p2}$
($s = \log(CS_{dp2}/CS_{dp1})/\log(d_{p2}/d_{p1})$). The value of $s$ can be derived from the measured particle size
distribution and was found to be around -1.6 for the present study, which is in good agreement with the
values reported by Lehtinen et al. (2007). The growth rate was derived from the nDMA measurements
in the size range between 3 and 10 nm by fitting a Gaussian function to the particle size distribution to
determine the mode diameter for all measured size distributions. Applying a linear fit to the mode
diameter as a function of time yields the $GR$ used in equation (8) (Hirsikko et al., 2005). Errors are
calculated by taking into account the statistical variation of the particle formation rates $J_{dp2}$ as well as
systematic errors on $GR$ (factor of 2), $d_{p2}$ (factor 1.3) and $CS$ (factor 1.5).
Figure 9 shows a comparison between $J_{dp1}$ from this study, data from other field studies and formation
rates from CLOUD chamber studies for the system of sulfuric acid, dimethylamine and water at 278 K
(Almeida et al., 2013) as well as for oxidized organic compounds with sulfuric acid and water
(Riccobono et al., 2014).


**4. Discussion**

By comparing time periods where significant new particle formation (NPF) occurred to time periods
where no NPF was observed, some conclusions can be drawn about the relevance of certain parameters
regarding NPF. Figure 10 shows a comparison for a variety of parameters by comparing nucleation days
to no nucleation days (red bars) and periods with high sulfuric acid dimer concentrations to no nucleation
days when there are also no high dimer concentrations (blue bars).
It is evident from Fig. 10 that sulfuric acid is on average a factor of 2 to 2.5 higher on days with
nucleation; although the variability is rather high (error bars take into account the standard deviations
of a parameter both for the nucleation days and the no nucleation days). The enhanced sulfuric acid
concentrations confirm the importance of this compound regarding NPF, which has also been shown in
numerous other studies (e.g. Weber et al., 1997; Kulmala et al., 2004; Fiedler et al., 2005; Kuang et al.,
2008). The OH concentration and the global radiation are also enhanced during nucleation, which is not
surprising given the fact that the parameters $H_2SO_4$, OH and global radiation are connected. The relative
humidity is generally lower during nucleation periods, which has also been reported in previous studies
(Hamed et al., 2011; Nieminen et al., 2015).
Regarding amines and ammonia Fig. 10 reveals an anti-correlation between their concentration and
the occurrence of NPF or sulfuric acid dimer formation (factor 2 to 5 lower during nucleation). However,
this does not necessarily mean that these compounds inhibit the formation of particles. On the contrary,
it could mean that amines and ammonia are efficiently taken up by small clusters and therefore are also
involved in the formation of new particles. Unlike sulfuric acid, amines and ammonia are not produced
in the gas phase and therefore their concentration will decrease with increasing distance from their
sources depending on the condensation sink. During nucleation the condensation sink is slightly
enhanced (Fig. 10), probably because of the newly formed particles. However, the $CS$ is only calculated
for particles larger than 3 nm. Also smaller particles and sulfuric acid clusters can contain amines
(Kürten et al., 2014) and even the sulfuric acid monomer can be bound to dimethylamine (Ortega et al.,
2012; Kürten et al., 2014). Therefore, continuous production of sulfuric acid and its clusters will lead to
a depletion of amines away from their sources, although no mixed clusters of sulfuric acid and amines
could be observed; this is probably the case because their concentrations were too low to be measured
with the CI-APi-TOF. As sulfuric acid concentrations are high during nucleation this could explain the
low amine values. Efficient uptake of amines in the particle phase has also been reported in a previous
field study (You et al., 2014). In addition, the limited pool of amines can also be the explanation for the
relatively low slope from Fig. 6 (sulfuric acid dimer vs. monomer) for some of the periods with elevated
sulfuric acid dimer concentrations. If the sulfuric acid concentration increases, the ratio of the free
(unbound) amine to sulfuric acid concentration drops, and there are fewer amines available to stabilize
the sulfuric acid dimers. This is a different situation compared to the CLOUD experiment where the
amine to sulfuric acid concentration was maintained at a ratio of ~100 over the entire duration of the
experiments. However, from these observations we cannot unambiguously conclude if the amines are
involved in the very first steps of nucleation, or if they are depleted due to clusters, which do not need
the help of amines in order to nucleate. One other aspect that could explain the low amine ratios is the
somewhat enhanced OH concentration during the nucleation days, as amines react with OH. However,
the life-time of amines regarding their reactions with OH is on the order of hours (Ge et al., 2011),
whereas the uptake on particles is significantly faster (if $CS$ is on the order of $10^{-3}$ to $10^{-2}$ s$^{-1}$).
Regarding the possibility that sulfuric acid and amines can explain the observed nucleation it has to
be noted that no clusters involving more than two sulfuric acid molecules could be observed. In the
following we will calculate the maximum expected sulfuric acid trimer concentration and discuss what
parameters can lower this concentration. The maximum measured sulfuric acid dimer concentration is
around $1 \times 10^5$ molecule cm$^{-3}$ for a sulfuric acid monomer concentration of $1 \times 10^7$ molecule cm$^{-3}$. A
sulfuric acid trimer will be formed through the collision between a monomer and a dimer (collision rate
$K_{1,2}$), whereas the loss rate of the trimer is defined by the sum of the condensation sink ($CS$) and its
evaporation rate ($k_{e,3}$). At steady-state this would yield the following equation for the trimer
concentration $N_3$ as function of the monomer and dimer concentrations $N_1$ and $N_2$ (for simplicity this
neglects a potential contribution from tetramer evaporation):

$$N_3 = \frac{K_{1,2} \cdot N_1 \cdot N_2}{CS + k_{e,3}}.$$    (10)

Using a value of $5 \times 10^{-10}$ molecule$^{-1}$ cm$^3$ s$^{-1}$ for $K_{1,2}$ and a condensation sink ($CS$) of $5 \times 10^{-3}$ s$^{-1}$ for the
above mentioned monomer and dimer concentrations would yield a trimer concentration of $1 \times 10^5$
molecule cm$^{-3}$ if the trimer evaporation rate would be zero. This concentration should be detectable with
our CI-APi-TOF. The fact that we do not see the trimer could indicate that the trimer evaporation rate
is non-zero. For a high amine to sulfuric acid ratio nucleation proceeds at or close to the kinetic limit
(Jen et al., 2014; Kürten et al., 2014). However, if the amine concentration is not very high, not every
trimer that is formed would be stable (as it is the case for a favored acid-base ratio, see Ortega et al.,
2012) and therefore could evaporate rapidly. This would result in lower trimer concentrations, which
could be below the detection limit of the CI-APi-TOF. From this perspective the absence of larger
sulfuric acid amine clusters is not necessarily an indication that this system is not responsible for new
particle formation. In other regions where the sulfuric acid and amine mixing ratios are even higher (i.e.
very close to amine sources) such clusters can be observable (Zhao et al., 2011). Recently, Jen et al.
(2016b) provided evidence that nitrate chemical ionization could not be sensitive towards sulfuric acid-
amine or sulfuric acid-diamine clusters if these contain three or more sulfuric acid molecules because
of the lowered acidity of such clusters by the basic amines/diamines. This could also explain the absence
of clusters beyond the dimer in the present study. Further measurements using different primary ions
are needed to investigate this possibility further.
The C10 and C20 signals for NPF and no nucleation days are not significantly different (Fig. 10).
This can be interpreted in different ways: (1) the HOM are not important in terms of NPF, (2) HOM are
generally high enough and it needs just enough sulfuric acid to initiate nucleation involving HOM, or
(3) „HOM" is too broadly defined and only a subgroup of HOM is involved in the nucleation but
currently we cannot distinguish this group. Neither of the possibilities can be proven right or wrong.
However, what can be said is that it is unlikely that the identified HOM alone are capable of producing
new particles to a significant extent at the conditions of the present study. The HOM dimer
concentrations (Fig. 8) are significantly higher during the night than during the day. Nevertheless, no
night-time nucleation is observed. This could be interpreted as an indication that if HOM are involved
in NPF it requires additional compounds such as sulfuric acid to initiate significant nucleation.
Alternative explanations for the absence of night time nucleation could be the suppression of the
formation of HOM that can nucleate by NO$_3$ during the night, or low [OH], which is required for the
formation of nucleating HOM.
Kulmala et al. (2013) proposed that $C_{10}H_{15}NO_8$ (detected as a cluster with NO$_3^-$ at 339.0681 Th)
could be important because NPF correlated even better with this compound compared to sulfuric acid.
During nucleation days this compound is only slightly elevated (Fig. 10) and this could be due to the
generally higher OH levels although the exact formation mechanism of $C_{10}H_{15}NO_8$ has to our knowledge
not been reported yet. During nucleation, no mixed clusters between sulfuric acid and HOM could be
identified. However, this also does not rule out their existence as the concentrations could be below the

CI-APi-TOF detection limit, or a low charging efficiency with the nitrate primary ion could prevent their detection. Furthermore, not all signals are identified yet.

The observed particle formation rates (Fig. 9) are consistent with the rates observed at other sites, although being at the upper end of the typical ranges that have been previously measured. The present data seem to agree a bit better to CLOUD chamber data for the system of sulfuric acid, water and dimethylamine (Almeida et al., 2013) compared to data for the system of sulfuric acid, water and oxidized organics from pinanediol (Riccobono et al., 2014). However, a direct comparison is difficult as the conditions between this ambient study and the CLOUD chamber experiments are not identical (with respect to $T$, $RH$, $CS$, amine mixing ratios, HOM concentrations, etc.).

**5. Summary**

In spring 2014 (May 18 to June 7) a field campaign was conducted at a rural site in central Germany (Vielbrunn/Odenwald). The measurement site was in proximity (within 450 to 1100 m distance) of three larger dairy farms. The perspective of this campaign was to evaluate if there is a connection between new particle formation and the concentration of amines and/or ammonia. Furthermore, the impact of highly oxidized organic molecules (HOM) from surrounding forests was investigated. A nitrate Chemical Ionization-Atmospheric Pressure interface-Time Of Flight mass spectrometer (CI-APi-TOF) was used to identify gas-phase compounds and clusters. Particle counters and differential mobility analyzers were used to characterize the aerosol size distribution and number density. The following conclusions can be drawn from our measurements:

- Nitrate CI-APi-TOF can be used to measure sulfuric acid, iodic acid, amines, a nitrosamine, ammonia and HOM; the measurement of iodic acid, ammonia and the nitrosamine has not been described before; the method is therefore even more versatile than previously thought and well suited to study all of the above-mentioned compounds during field measurements.

- The sulfuric acid concentration can be well described by proxies ($SO_2$, global radiation, RH and $CS$ or just by $SO_2$ and global radiation) for this site with a similar accuracy as reported in a previous study (Mikkonen et al., 2011).

- Significant sulfuric acid dimer concentrations were measured; it is, however, not clear what compound stabilizes the neutral dimers; larger sulfuric acid clusters (trimer and beyond) were not observed.

- Amines (C1-, C2-, C3-, C4- and C6-amines) are present at estimated mixing ratios between approximately 1 and 5 pptv, which is consistent with other studies; the C4- and C6-amines as well as ammonia show a diurnal variation, which follows the temperature profile.

- Iodine has been observed (probably iodic acid) on every day, somewhat surprising for a continental site located more than 400 km away from the ocean; the nitrate CI-APi-TOF has a

high sensitivity towards iodic acid and its presence indicates long-range transport of iodine
containing substances (although a local source cannot entirely be ruled out); using OH
concentrations also OIO concentrations can be estimated; however, both $[HIO_3]$ ($\sim3\times10^5$
molecule $cm^{-3}$) and $[OIO]$ ($\sim5\times10^6$ molecule $cm^{-3}$) are probably too low to affect new particle
formation significantly at this site.
• The diurnal pattern of HOM dimers shows maximum concentrations during the night but no night
time nucleation is observed; the day time concentration could be low due to the presence of NO
and/or $HO_2$ which suppress the HOM dimer formation.
• Relatively high particle formation rates are found, which are rather at the upper end of the
atmospheric observations for other rural sites; the rates are compatible with CLOUD chamber
data both for the systems of sulfuric acid, water and dimethylamine (Almeida et al., 2013), as well
as for a system involving sulfuric acid, water and oxidized organics (Riccobono et al., 2014); no
definitive answer can be given which system is more relevant.
• Nucleation seems to be favored on days with relatively low RH and high sulfuric acid; an anti-
correlation with the amine and ammonia signals is observed, this could be due to efficient uptake
of these compounds on clusters and particles during NPF as amines and ammonia are not
produced in the gas-phase.
The above bullet points seem to support recent findings about the relevance of amines in terms of NPF
and early growth (Chen et al., 2012; Almeida et al., 2013; Kulmala et al., 2013; Lehtipalo et al., 2016).
However, it cannot be unambiguously concluded that amines are more relevant for NPF than HOM at
this site because no nucleating clusters could be directly observed. More studies like the present one are
necessary in the future to obtain better statistics about the parameters relevant for NPF (Fig. 10). Ideally,
such measurements should include further instrumentation including a PSM (Vanhanen et al., 2011) for
the measurement of clusters and small particles (< 3 nm), an APi-TOF (Junninen et al., 2010) for
identification of charged nucleating clusters, an instrument for $HO_x/RO_x$ measurements (Mauldin et al.,
2016) and an instrument for sensitive amine measurements capable of speciating the amines.
**Acknowledgments**

We thank the German Weather Service (Deutscher Wetterdienst, DWD) for providing infrastructure and
meteorological data. Funding from the German Federal Ministry of Education and Research (grant no.
01LK1222A) and the Marie Curie Initial Training Network "CLOUD-TRAIN" (grant no. 316662) is
gratefully acknowledged.

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

**Table 1.** List of ions used in the identification of sulfuric acid (monomer and dimer), iodic acid,
ammonia, amines (C1, C2, C3, C4 and C6) and dimethylnitrosamine. Cattle farms in the vicinity of the
measurement site are expected to be a source for the listed amines (Ge et al., 2011).

| Ion | Exact mass | Neutral compound |
|---|---|---|
| $HSO_4^-$, $(HNO_3)HSO_4^-$ | 96.9601, 159.9557 | sulfuric acid monomer |
| $(H_2SO_4)HSO_4^-$, $(HNO_3)(H_2SO_4)HSO_4^-$ | 194.9275, 257.9231 | sulfuric acid dimer |
| $IO_3^-$, $(H_2O)IO_3^-$, $(HNO_3)IO_3^-$ | 174.8898, 192.9003, 237.8854 | iodic acid |
| $(NH_3)(HNO_3)_{1,2}NO_3^-$ | 142.0106, 205.0062 | ammonia |
| $(CH_5N)(HNO_3)_{1,2}NO_3^-$ | 156.0262, 219.0219 | C1-amines (e.g. methylamine) |
| $(C_2H_7N)(HNO_3)_{1,2}NO_3^-$ | 170.0419, 233.0375 | C2-amines (e.g. ethylamine, dimethylamine) |
| $(C_3H_9N)(HNO_3)_{1,2}NO_3^-$ | 184.0575, 247.0532 | C3-amines (e.g. trimethylamine, propylamine) |
| $(C_4H_{11}N)(HNO_3)_{1,2}NO_3^-$ | 198.0732, 261.0688 | C4-amines (e.g. diethylamine, butylamine) |
| $(C_6H_{15}N)(HNO_3)_{1,2}NO_3^-$ | 226.1045, 289.1001 | C6-amines (e.g. triethylamine) |
| $(C_2H_6N_2O)(HNO_3)_{1,2}NO_3^-$ | 199.0320, 262.0277 | dimethylnitrosamine |


**Table 2.** Peak list of the highly oxidized organic molecules (HOM) used in this study.

| Ion sum formula | Cluster ion | Exact mass | Compound |
|---|---|---|---|
| $C_{10}H_{15}NO_9^-$ | $(C_{10}H_{15}O_6)NO_3^-$ | 293.0752 | HOM $RO_2$ radical |
| $C_{10}H_{15}NO_{10}^-$ | $(C_{10}H_{15}O_7)NO_3^-$ | 309.0701 | HOM $RO_2$ radical |
| $C_{10}H_{15}NO_{11}^-$ | $(C_{10}H_{15}O_8)NO_3^-$ | 325.0651 | HOM $RO_2$ radical |
| $C_{10}H_{15}NO_{12}^-$ | $(C_{10}H_{15}O_9)NO_3^-$ | 341.0600 | HOM $RO_2$ radical |
| $C_{10}H_{15}NO_{13}^-$ | $(C_{10}H_{15}O_{10})NO_3^-$ | 357.0549 | HOM $RO_2$ radical |
| $C_{10}H_{15}NO_{15}^-$ | $(C_{10}H_{15}O_{12})NO_3^-$ | 389.0447 | HOM $RO_2$ radical |
| $C_8H_{12}NO_{11}^-$ | $(C_8H_{12}O_8)NO_3^-$ | 298.0416 | HOM monomer |
| $C_9H_{14}NO_{12}^-$ | $(C_9H_{14}O_9)NO_3^-$ | 328.0521 | HOM monomer |
| $C_{10}H_{14}NO_{10}^-$ | $(C_{10}H_{14}O_7)NO_3^-$ | 308.0623 | HOM monomer |
| $C_{10}H_{14}NO_{11}^-$ | $(C_{10}H_{14}O_8)NO_3^-$ | 324.0572 | HOM monomer |
| $C_{10}H_{14}NO_{12}^-$ | $(C_{10}H_{14}O_9)NO_3^-$ | 340.0521 | HOM monomer |
| $C_{10}H_{14}NO_{13}^-$ | $(C_{10}H_{14}O_{10})NO_3^-$ | 356.0471 | HOM monomer |
| $C_{10}H_{14}NO_{14}^-$ | $(C_{10}H_{14}O_{11})NO_3^-$ | 372.0420 | HOM monomer |
| $C_{10}H_{16}NO_9^-$ | $(C_{10}H_{16}O_6)NO_3^-$ | 294.0831 | HOM monomer |
| $C_{10}H_{16}NO_{10}^-$ | $(C_{10}H_{16}O_7)NO_3^-$ | 310.0780 | HOM monomer |
| $C_{10}H_{16}NO_{11}^-$ | $(C_{10}H_{16}O_8)NO_3^-$ | 326.0729 | HOM monomer |
| $C_{10}H_{16}NO_{12}^-$ | $(C_{10}H_{16}O_9)NO_3^-$ | 342.0678 | HOM monomer |
| $C_{10}H_{16}NO_{13}^-$ | $(C_{10}H_{16}O_{10})NO_3^-$ | 358.0627 | HOM monomer |
| $C_{10}H_{16}NO_{14}^-$ | $(C_{10}H_{16}O_{11})NO_3^-$ | 374.0576 | HOM monomer |
| $C_{10}H_{15}N_2O_{10}^-$ | $(C_{10}H_{15}NO_7)NO_3^-$ | 323.0732 | HOM nitrate |
| $C_{10}H_{15}N_2O_{11}^-$ | $(C_{10}H_{15}NO_8)NO_3^-$ | 339.0681 | HOM nitrate |
| $C_{10}H_{15}N_2O_{12}^-$ | $(C_{10}H_{15}NO_9)NO_3^-$ | 355.0630 | HOM nitrate |
| $C_{10}H_{15}N_2O_{13}^-$ | $(C_{10}H_{15}NO_{10})NO_3^-$ | 371.0580 | HOM nitrate |
| $C_{10}H_{15}N_2O_{14}^-$ | $(C_{10}H_{15}NO_{11})NO_3^-$ | 387.0529 | HOM nitrate |
| $C_{10}H_{15}N_2O_{15}^-$ | $(C_{10}H_{15}NO_{12})NO_3^-$ | 403.0478 | HOM nitrate |
| $C_{10}H_{15}N_2O_{16}^-$ | $(C_{10}H_{15}NO_{13})NO_3^-$ | 419.0427 | HOM nitrate |
| $C_{10}H_{16}N_3O_{11}^-$ | $(C_{10}H_{16}N_2O_8)NO_3^-$ | 354.0790 | HOM di-nitrate |
| $C_{10}H_{17}N_4O_{14}^-$ | $(C_{10}H_{16}N_2O_8)(HNO_3)NO_3^-$ | 417.0747 | HOM di-nitrate |
| $C_{10}H_{16}N_3O_{12}^-$ | $(C_{10}H_{16}N_2O_9)NO_3^-$ | 370.0739 | HOM di-nitrate |
| $C_{10}H_{17}N_4O_{15}^-$ | $(C_{10}H_{16}N_2O_9)(HNO_3)NO_3^-$ | 433.0696 | HOM di-nitrate |
| $C_{10}H_{16}N_3O_{13}^-$ | $(C_{10}H_{16}N_2O_{10})NO_3^-$ | 386.0689 | HOM di-nitrate |
| $C_{10}H_{17}N_4O_{16}^-$ | $(C_{10}H_{16}N_2O_{10})(HNO_3)NO_3^-$ | 449.0645 | HOM di-nitrate |
| $C_{19}H_{30}NO_{16}^-$ | $(C_{19}H_{30}O_{13})NO_3^-$ | 528.1570 | HOM dimer |
| $C_{19}H_{30}NO_{17}^-$ | $(C_{19}H_{30}O_{14})NO_3^-$ | 544.1519 | HOM dimer |
| $C_{20}H_{28}NO_{16}^-$ | $(C_{20}H_{28}O_{13})NO_3^-$ | 538.1414 | HOM dimer |
| $C_{20}H_{28}NO_{17}^-$ | $(C_{20}H_{28}O_{14})NO_3^-$ | 554.1363 | HOM dimer |
| $C_{20}H_{28}NO_{18}^-$ | $(C_{20}H_{28}O_{15})NO_3^-$ | 570.1312 | HOM dimer |
| $C_{20}H_{28}NO_{19}^-$ | $(C_{20}H_{28}O_{16})NO_3^-$ | 586.1261 | HOM dimer |
| $C_{20}H_{28}NO_{20}^-$ | $(C_{20}H_{28}O_{17})NO_3^-$ | 602.1210 | HOM dimer |
| $C_{20}H_{28}NO_{21}^-$ | $(C_{20}H_{28}O_{18})NO_3^-$ | 618.1159 | HOM dimer |
| $C_{20}H_{28}NO_{22}^-$ | $(C_{20}H_{28}O_{19})NO_3^-$ | 634.1108 | HOM dimer |
| $C_{20}H_{28}NO_{23}^-$ | $(C_{20}H_{28}O_{20})NO_3^-$ | 650.1058 | HOM dimer |
| $C_{20}H_{30}NO_{17}^-$ | $(C_{20}H_{30}O_{14})NO_3^-$ | 556.1519 | HOM dimer |


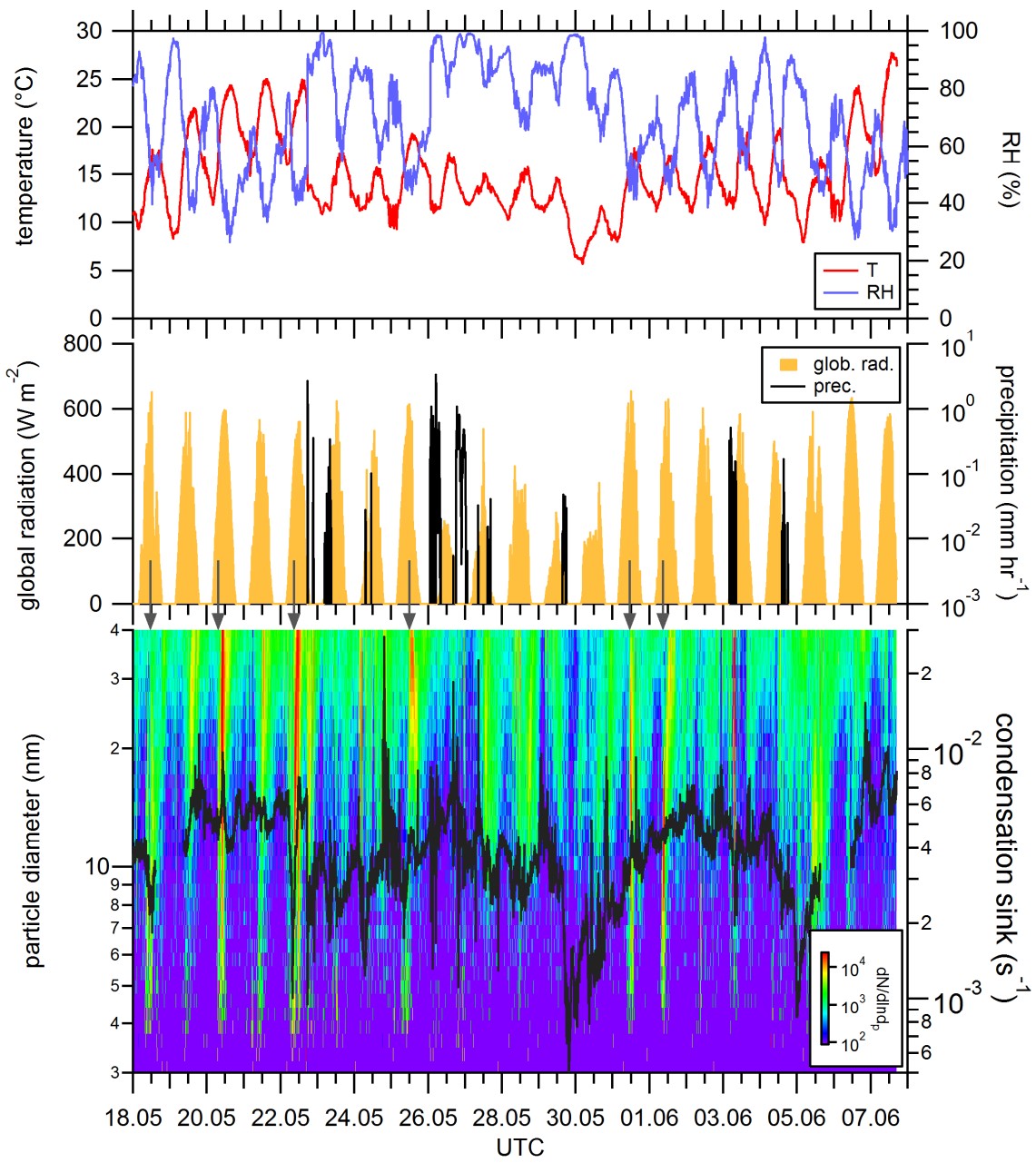

**Fig. 1.** Overview of the different parameters measured between May 18 and June 8, 2014. Temperature (T) and relative humidity (RH) are shown in the upper panel, the center panel shows the global radiation and precipitation, while the bottom panel shows the number size distribution measured by the nano-DMA together with the condensation sink (black line). Grey arrows above the bottom panel indicate days when significant NPF was observed.

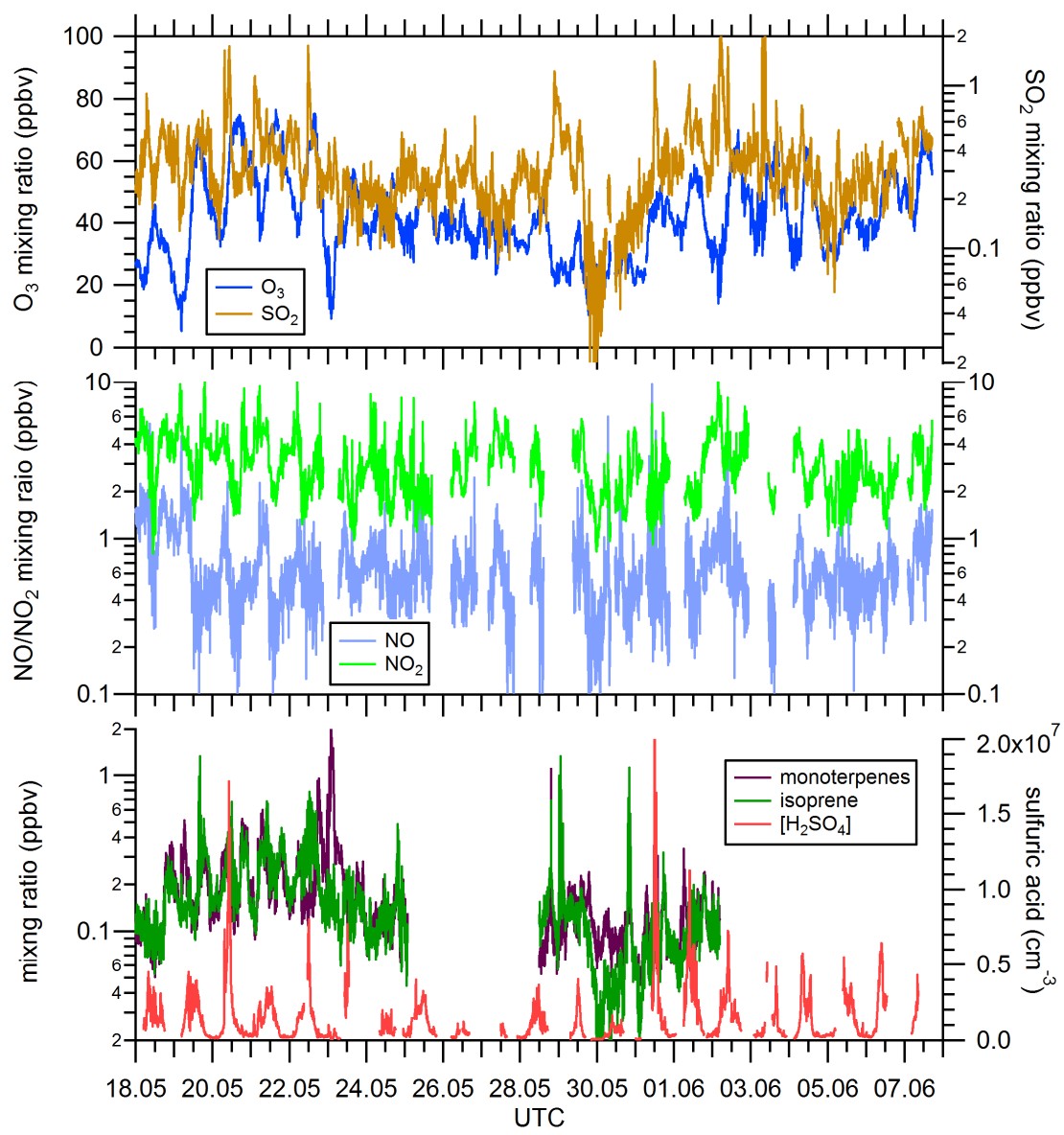

1282

**Fig. 2.** Overview of the trace gas measurements between May 18 and June 8, 2014. The $SO_2$ and $O_3$
mixing ratios are shown in the upper panel, the NO and $NO_2$ mixing ratios are shown in the center panel
and the bottom panel shows the sulfuric acid monomer concentration ($[H_2SO_4]$) together with the
isoprene and monoterpene mixing ratios.

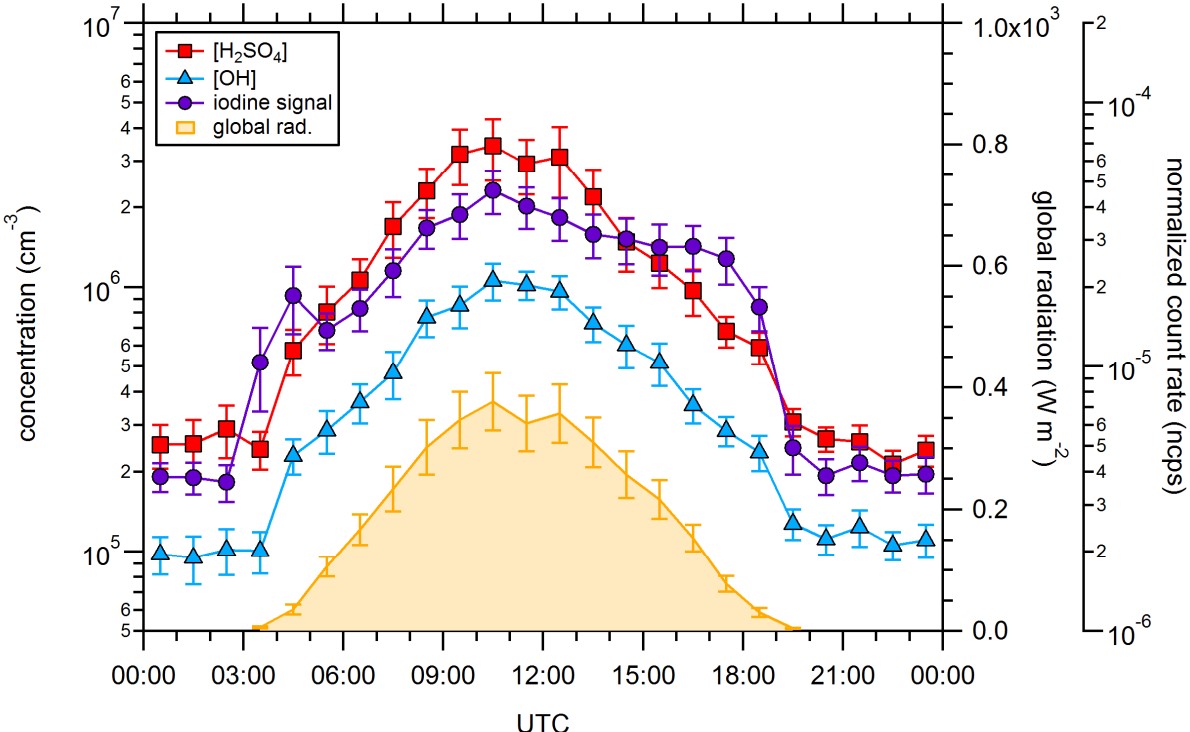

**Fig. 3.** Diurnal averages for the sulfuric acid ([H$_2$SO$_4$]) and the calculated hydroxyl radical ([OH]) concentrations (axis on the left). The iodine signal is not converted into a concentration, instead the normalized count rates per second (ncps) are shown (axis on the right). A value of $5\times10^{-5}$ ncps for iodine would correspond to a concentration of $3\times10^5$ molecule cm$^{-3}$ applying the same conversion factor for iodic acid as for sulfuric acid. The global radiation indicates that all signals are related to photochemistry. Error bars indicate one standard deviation for the 30-minute averaged values.

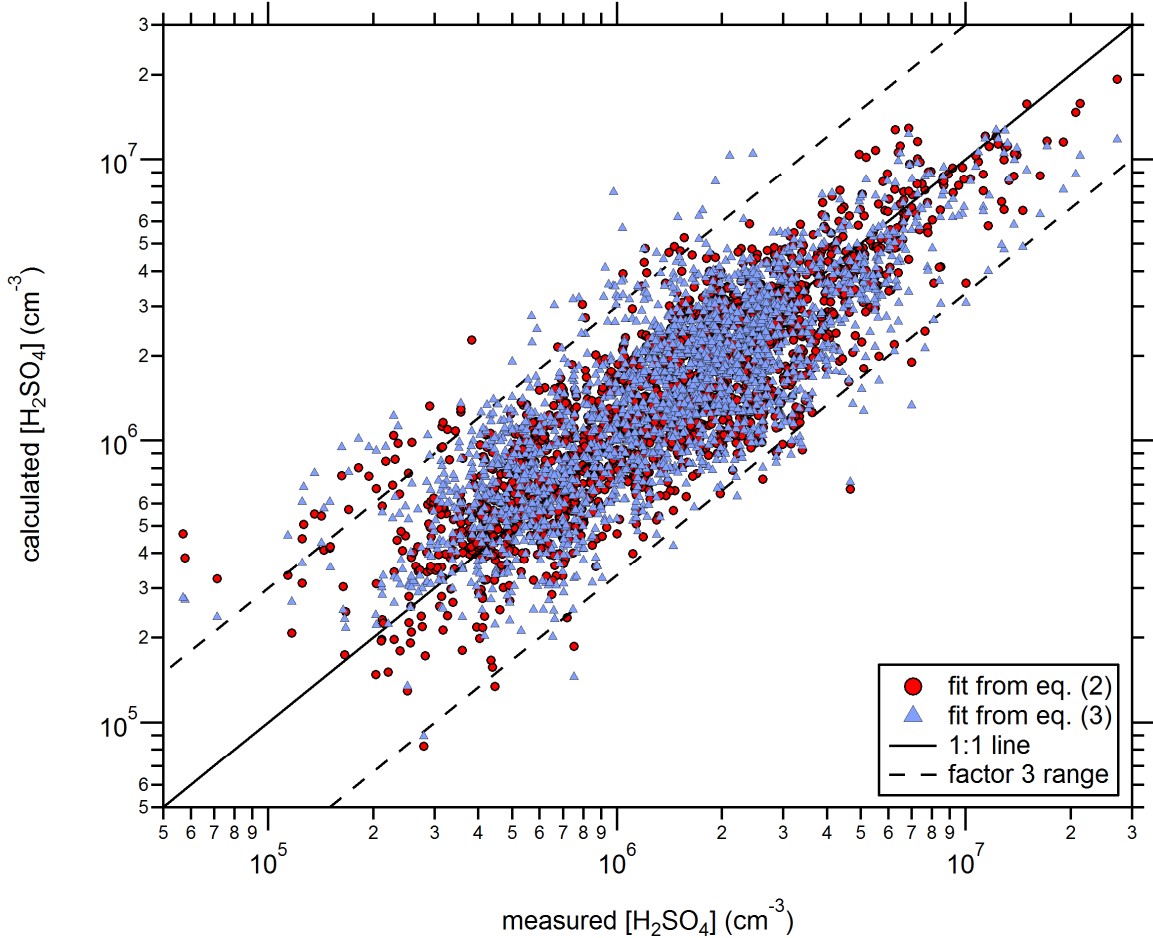

**Fig. 4.** Calculated [$H_2SO_4$] as a function of the measured concentrations. Only data points where the global radiation exceeded 50 W m$^{-2}$ were considered in deriving the fit parameters for equations (2) and (3). The red circles take into account $SO_2$, global radiation (*Rad*), condensation sink (*CS*) and relative humidity (RH) to calculate the [$H_2SO_4$], whereas only $SO_2$ and global radiation are used for the blue triangles.

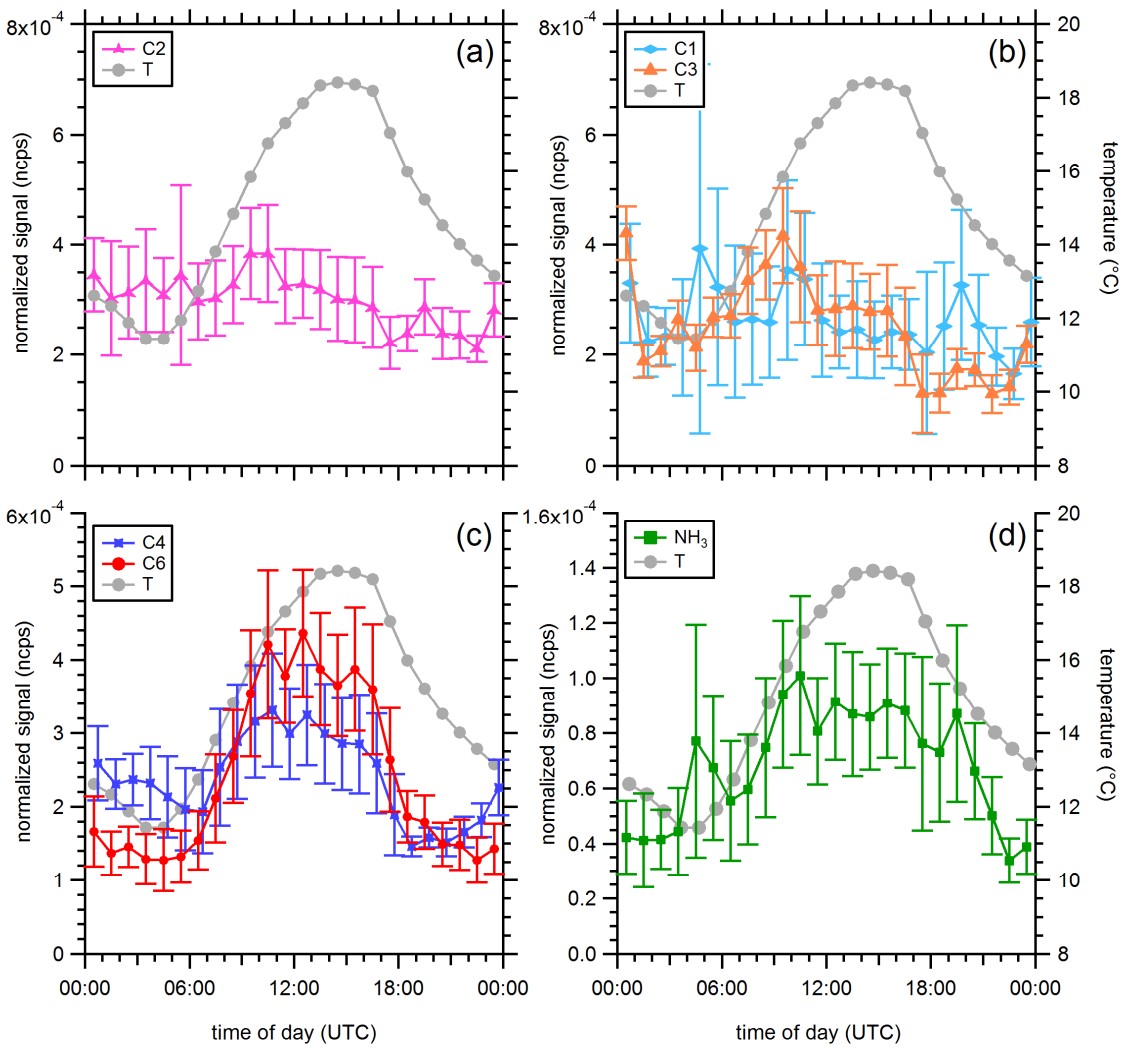

1300

**Fig. 5.** Diurnal averages for different amines (C1, C2, C3, C4 and C6) and ammonia. The temperature
profile is shown in addition. Error bars represent one standard deviation of the 30-minute averages. The
lower detection limits for the different compounds are not well-defined, however, the lowest measured
signals during some periods were $0.3 \times 10^{-4}$ ncps for C1, $\sim 0.5 \times 10^{-4}$ ncps for C2, C3, C4 and C6 and
$0.1 \times 10^{-4}$ ncps for ammonia. For most of the time (and for all averaged values shown) the signals were
clearly above these "background" levels.

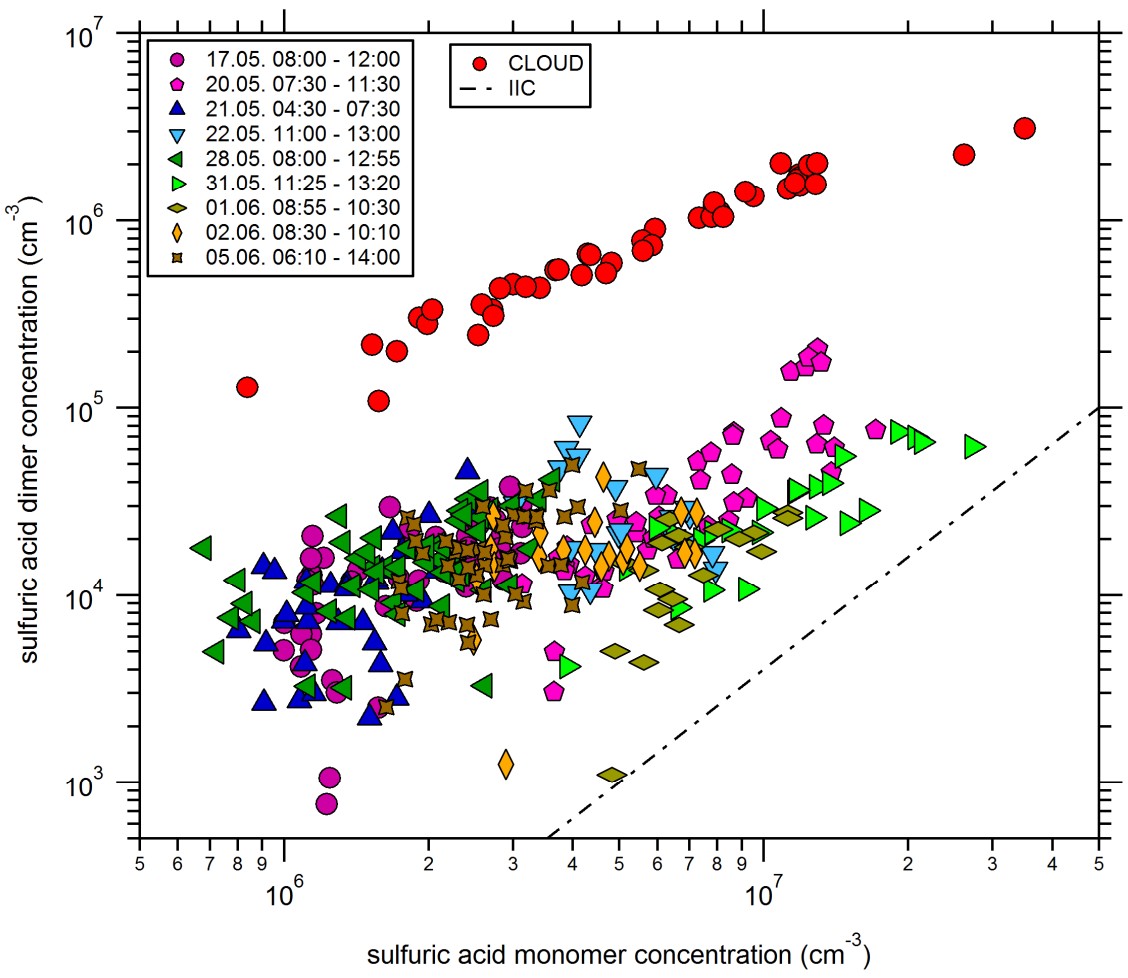

**Fig. 6.** Sulfuric acid dimer concentrations as a function of the sulfuric acid monomer concentrations. The legend on the left lists the periods when high dimer signals were observed. In addition, data from CLOUD chamber experiments with at least 10 pptv of dimethylamine are shown; under these conditions dimer formation proceeds at or close to the kinetic limit (Kürten et al., 2014). The dashed-dotted line indicates the lower detection limit for neutral dimers set by ion-induced clustering (IIC) within the CI-APi-TOF ion reaction zone.

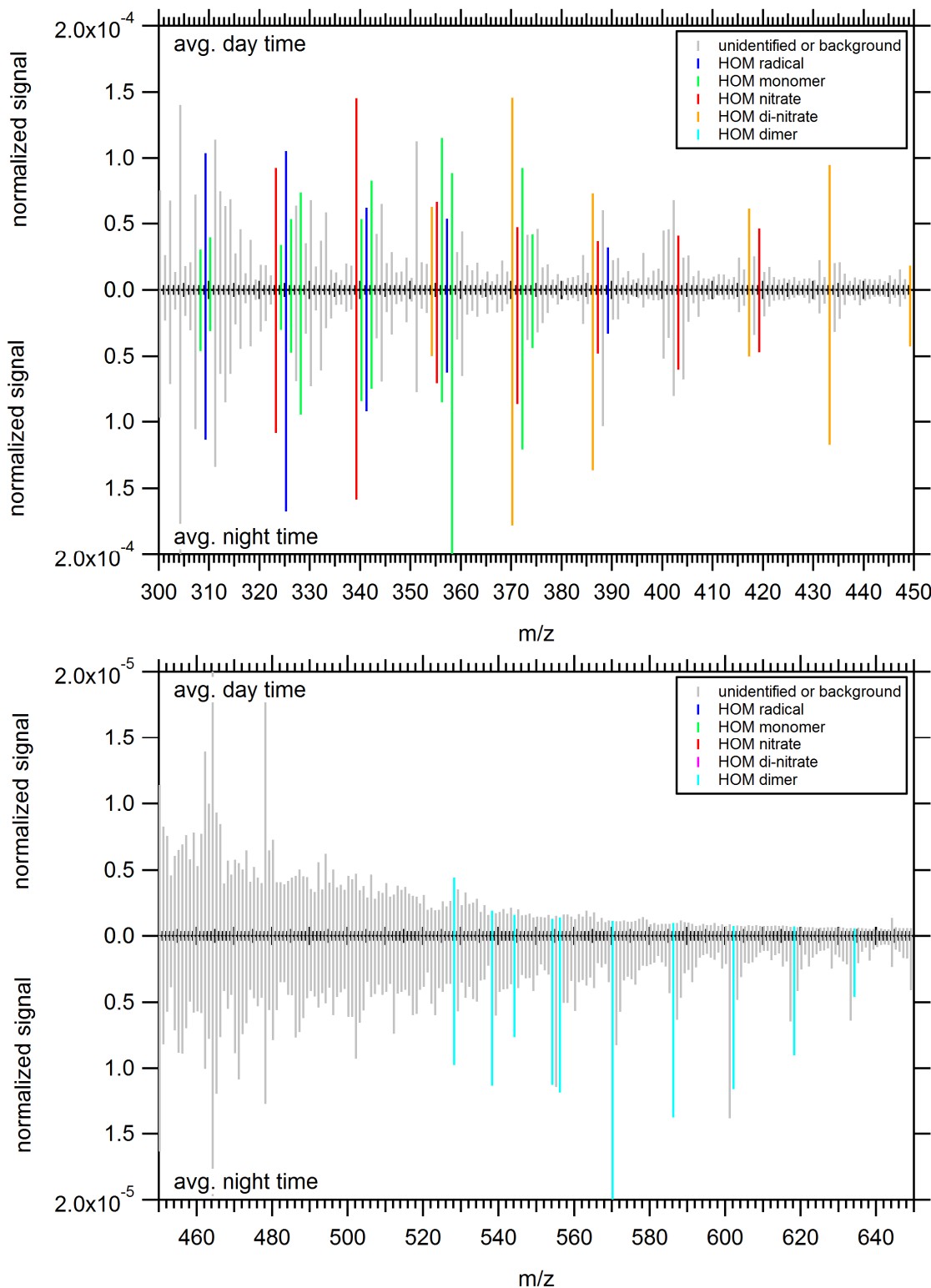

1314

**Fig. 7.** Comparison between average day time and night time mass spectra measured with the nitrate CI-
APi-TOF. The day time spectrum was averaged for periods when no nucleation was observed.

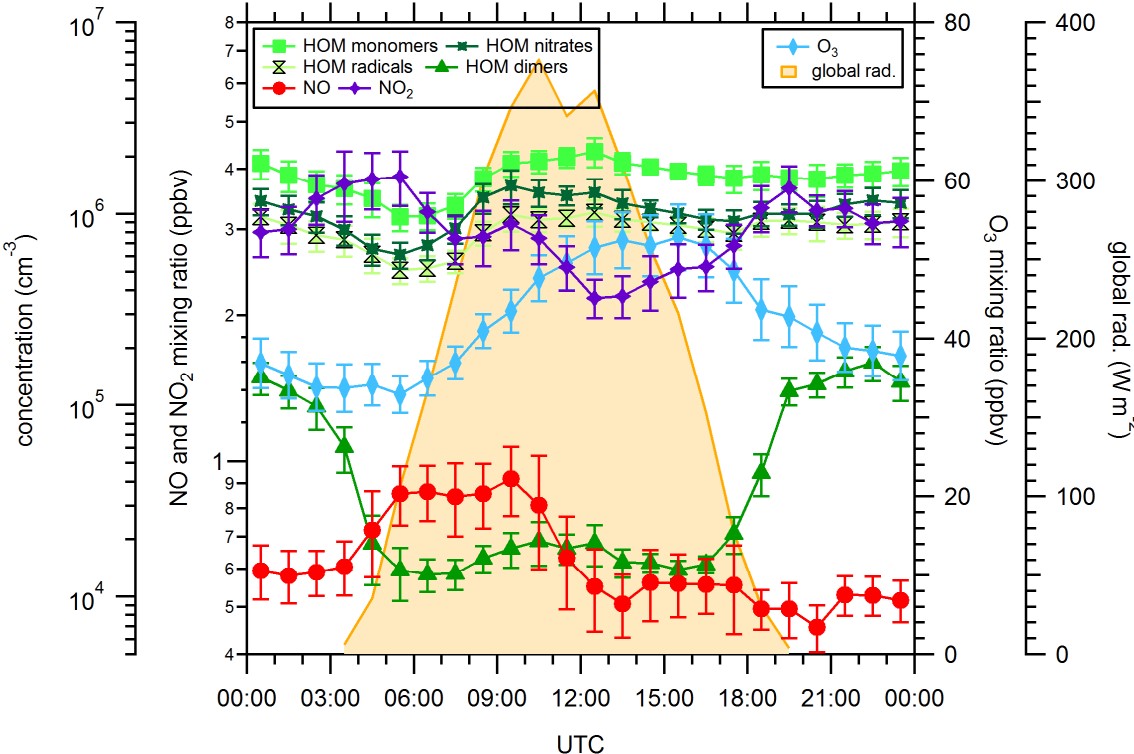

1317

**Fig. 8.** Diurnal profiles of the NO, $NO_2$ and $O_3$ mixing ratios. The signals for highly oxidized organic molecules (HOM) are shown for some C10 (HOM monomers, HOM nitrates and HOM radicals) and C19/C20 compounds (HOM dimers), which show a distinct maximum during the night. The HOM dinitrates show a similar pattern as the other C10 compounds and are not included in the figure. The global radiation is shown in addition. Error bars indicate one standard deviation for the 30-minute averages.

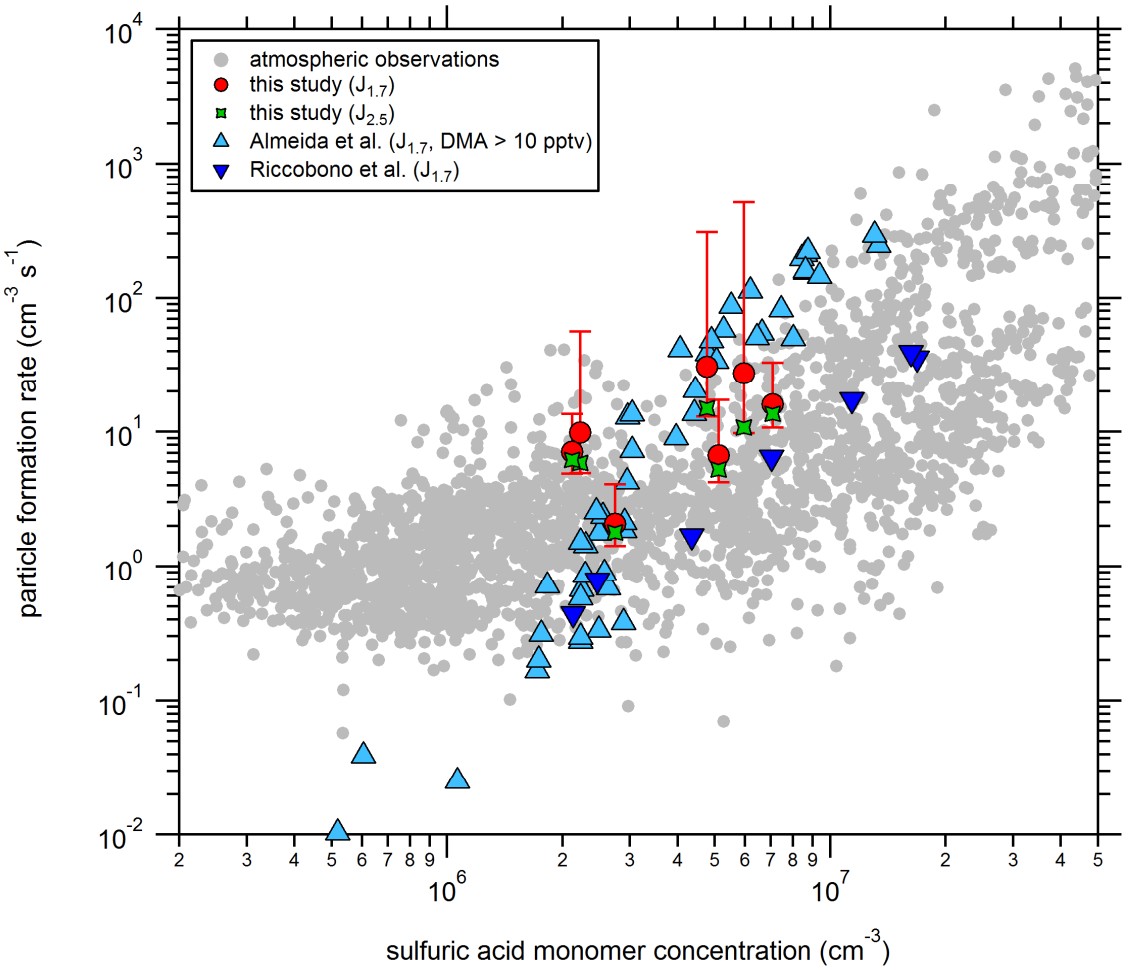

**Fig. 9.** Particle formation rates from this study at a mobility diameter of 1.7 nm ($J_{1.7}$ red circles) and 2.5 nm ($J_{2.5}$, green stars). Data from CLOUD chamber measurements for a diameter of 1.7 nm are shown in addition for the system of sulfuric acid, water and dimethylamine (light blue symbols, see Almeida et al., 2013) and sulfuric acid, water and oxidized organics from pinanediol (dark blue symbols, see Riccobono et al., 2014). The light grey circles are from other field measurements (Kuang et al., 2008; Paasonen et al., 2010; Kulmala et al., 2013).

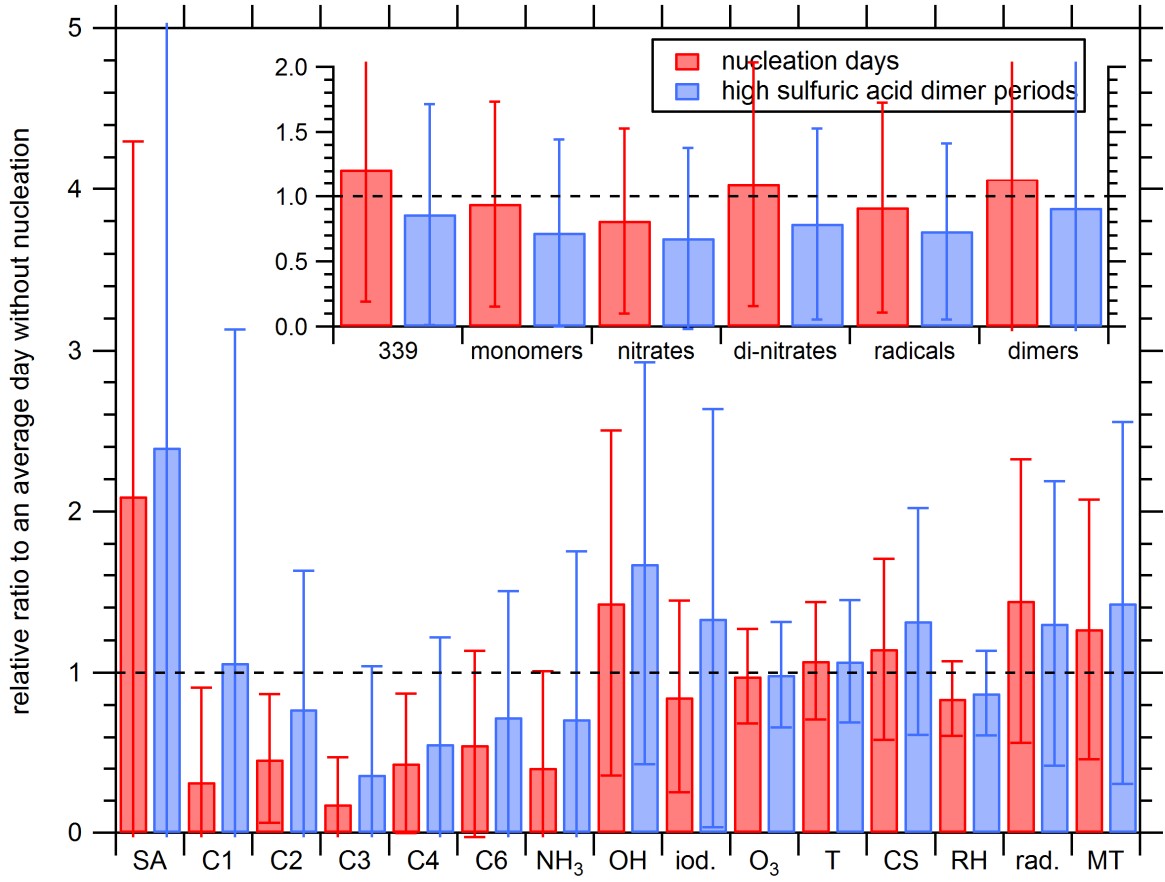

**Fig. 10.** Comparison of various parameters for different time periods (SA = sulfuric acid monomer, C1, C2, C3, C4 and C6 = amines, iod. = iodic acid and rad. = global radiation intensity). The subset figure on the upper right shows the signals for the highly oxidized organic compounds with 10 or 20 carbon atoms (339 = organic compound $C_{10}H_{15}NO_8$ clustered with $NO_3^-$ having a mass of 339.0681 Th, the definition of other HOM, i.e. monomers, radicals, nitrates, di-nitrates and dimers can be found in Table 2). The red bars relate nucleation days to days without nucleation and the blue bars show the ratio between periods where high sulfuric acid dimer concentrations were observed (see Fig. 6) to no nucleation days. Similar times of the day (early morning) were used as reference periods when no nucleation was observed as nucleation and dimer formation was also mainly observed in the morning.