# Peer review of "central Germany"

_Atmospheric Chemistry and Physics, 2016_

## Referee Comment (RC1) · Anonymous Referee #2 · 11 Jul 2016

Summary: The motivation for this publication was to use concurrent ambient measurements of ultrafine particles and gaseous precursors to particle formation as evidence to investigate the hypothesis that different gaseous species are more important than others for forming particles at this measurement site. Another important feature of this publication included the investigation of nitrate CIMS as a novel method for measuring ammonia, amines and iodine-containing compounds. Analysis of the variability of mechanisms controlling new particle formation in different areas where the chemical composition of the lower atmosphere is influenced heavily by unique emission sources, such as feedlots, is a pursuit with high scientific merit. The authors attempt to analyze

how different sources could be contributing to NPF well by exploiting the capabilities of the nitrate CIMS technique and considering contributions of HOMs to NPF potentially sourced from a local mixed forest. This manuscript is recommended for publication. Included are general comments, technical suggested corrections, and recommendations for improvements of analysis and the method of measurement for future studies.

Comments on analysis and suggestions for improvement:

One of the things that is pointed out in the introduction is that new measurement techniques for amines have appeared over the past few years and it is unclear how spatially variable amines are because it is uncertain how well these techniques actually, quantitatively, measure amines. I think future studies should include a more rigorous approach at an attempt to calibrate this technique to different amines. Attempts at a rigorous calibration of amines for this technique would help to understand, with greater confidence, the spatial variability in concentrations amines around the area of study in this publication, as well as, set a higher standard for measurement of these species by other users of this nitrate CIMS technique. A more rigorous approach to calibration would help advance the understanding of the abundance of atmospheric amines globally, in addition to advancing the understanding of how significant of a role atmospheric amines play in NPF.

Line 154 It would be helpful to add in chemical equations describing the ionization reactions thought to dominate ionization chemistry using the nitrate CIMS technique that are additions to the descriptions on line 154.

Line 170 "while assuming the same ionization efficiency as for sulfuric acid, which has been argued to be a valid assumption" . . . In the supplemental, or appendix?, of Ehn (2014) the authors present the results of quantum computations of binding energies and calculate theoretical collisional rate constants for ELVOCs that are within the measured error for sulfuric acid. Additionally, they compare these calculated rate constants to the measured rate constant based on the sensitivity of the nitrate CIMS

measurement to a perfluoroacid. This analysis presented in Ehn (2014), in my opinion, is really good because authentic standards of HOMs are not available. Additionally, because the quantum calculations used structure-activity relationships to derive theoretical dipole moments for some theoretical HOM structures, the results suggesting that clustering of HOMs occurs at the collisional limit only applies to HOMs. It seems one of the primary motivations for the publication of the mass-dependent transmission efficiency study by Heinritzi (2016) was to characterize these effects for HOMs. I believe quantification of amines using nitrate CIMS through direct calibrations would greatly improve the confidence in the measurement technique, and may even help further support previous attempts at quantifying HOMs with this technique if many directly calibrated compounds appear to undergo ionization at the collisional limit. The importance of direct calibrations is further underscored through the authors suggestion that the nitrate CIMS technique may be particularly less sensitive to ammonia than other compounds they are able to measure.

Line 296 An argument is made here that a simpler model can be used to estimate sulfuric acid concentration because the results from this model consistently agree with the results from a more detailed model, which includes RH and CS as variables, within a factor of three. Mikkonen (2011) also points out the minor dependence of the approximated sulfuric acid concentrations on CS. The authors suggest a simplified equation in the conclusions which is only dependent on radiation, the rate constant, and SO2 mixing ratio if particle data is not available. I would argue that if a modeled approximation, and not a measurement, is being used to represent data then it should be treated as rigorously as possible. Including the RH and CS as sinks for SO2, even if they contribute in minor ways, I think is important when forming this approximation, if the data is available. To constrain the approximated data by a factor of three, I think, is a worthwhile action that the authors of the current manuscript took when using the detailed sulfuric acid approximation equation.

The discussion section provides some constructive suggestions for improving future

measurements!

Technical suggestions: **Suggestions for technical corrections ARE IN ALL CAPS

Line 25 "WE DEMONSTRATE HERE that the nitrate CI-APi-TOF is suitable for sensitive measurements. . ."

Line 114 ". . .the site is not too far away 114 from the University of Frankfurt, which allowED US to visit the station for instrument maintenance" . . . That sentence either needs to be modified or changed by the above suggestion because right now, grammatically, it doesn't make any sense.

Line 121 "As mentioned in the introduction, livestock ARE known to emit a variety of amines as well as ammonia. . ."

Line 303 . . . Please make an explicit statement of the reasons that it was useful to estimate the OH radical concentration as the introduction to this section. This would help to disambiguate the meaning of "further data evaluation", but also help guide, and preview to, the reader to the logic of what the authors saw as the ultimate purpose of doing analyses using estimated OH radical concentrations. This should be a quick and easy modification.

Line 309 . . . Please explicitly define the quantity [OH]day. Adding the "day" notation to this quantity makes the exact meaning confusing. I believe this quantity just refers to the steady-state approximated concentration of OH radicals at some time that it is calculated. A similar quantity, OHday, is used in the PAM/OFR literature to refer to an estimate of the day-averaged expected OH exposure.

Line 339 . . . "by tentatively adopting the same calibration constant for iodic acid AS sulfuric acid"

Line 367 . . . I believe the idea reported by the authors that "iodic acid (and, if present, probably also its clusters) can be detected with high sensitivity due to the high negative mass defect of the iodine atom" is inaccurate. Sensitivity is formally defined as

the change in instrument response with respect to a change in some amount of analyte. The sensitivity of the CIMS measurement to any compound is a function of the ionization chemistry and ion transmission through the instrument. What the authors are describing here is the one of the reasons why iodide ionization is a valued as an ionization method in CIMS measurements. The high negative mass defect of iodine produces a characteristic pattern of appearance in the mass spectrum which is useful when trying to unambiguously identify peaks when performing high resolution peak fitting, but this feature doesn't have any reflection on the sensitivity of this particular CIMS measurement to iodic acid. If the authors are measuring the concentrations of iodic acid that they estimate to be measuring then nitrate CIMS is probably particularly sensitive to some iodine compounds, but the logic reported on line 367, I believe, is misleading.

Line 375 . . . "the same calibration constant for iodic acid AS sulfuric acid"

---

## Referee Comment (RC2) · Anonymous Referee #1 · 22 Sep 2016

Measurements of sulphuric acid, amines, ammonia and VOC oxidation products are reported in connection with observations of atmospheric new particle formation (NPF) at a rural site in Germany. Focus of the manuscript is on showing that the nitrate CI-APi-TOF instrument is capable of measuring in ambient conditions ammonia, amines and oxidation products of organic compounds. These have been identified in recent laboratory studies to enhance sulphuric acid–water nucleation rates to atmospheric levels, therefore detecting them in atmospheric measurements is highly relevant for the current nucleation research. The measurements of sulphuric acid are further evaluated by comparing to steady-state proxy concentration. Reporting the proxy coefficients for this

environment provides valuable information, since ambient measurements of sulphuric acid are rare, and the proxies have been widely used in different environments.

The CI-APi-TOF data is used to make comparisons between days with and without occurrince of NPF, in order to find out which precursor species affect NPF at this site. No definite participation of ammonia, amines or HOMs to nucleation at this site could be made, but the possible reasons for this are adequately discussed in the manuscript. Also comparisons to chamber measurements from the CLOUD experiment are made.

The manuscript is well suited for publication in Atmos. Chem. Phys. I have listed some minor comments and correction/clarification suggestions below (in addition to those made by the anonymous referee 2).

Minor and technical comments:

Line 35: ". . . a larger fraction . . ." should be " . . . a large fraction . . ."

Line 78: Please add the abbreviation HOM here, as it is used later in the text.

Lines 208–210: What are the detection limits for the SO2, O3 and NOx monitors? In section 3.2, the lowest SO2 concentrations of 0.05 ppb sound quite low for a standard SO2 monitor.

Line 202: Does this mean the reaction rate constants for the proton transfer reaction in the PTR-MS are similar for different monoterpenes, and therefore are detected with similar efficiency as alpha-pinene?

Lines 242–244: Please check whether it was 6 or 7 NPF days during the campaign. In Section 3.9 (line 590) it is said 7 events and also Fig 9 shows seven J values.

Line 600: Why is the condensational growth out from the 2.5–10 nm size range not considered in Equation 7? That is an additional loss term for particles in this size range, so the right hand side of Eq 7 should have an additional term GR/(7.5 nm)*N (see Kulmala et al. (Nature Protocols 7, 1651–1667, 2012), Equation 9).

Caption text of Figure 1: Please add a mention that the arrows in the bottom panel indicate NPF days. Also check whether there should be six or seven days marked as NPF days (Fig 9 shows J rates for seven days).
* * *

---

## Author Comment (AC1) · 26 Sep 2016

We thank the referee for the constructive comments, which are added in full below (in black font). Our replies are given in blue font directly after the comments, text that has been added to the manuscript is shown in red font.

Referee #2:

Summary: The motivation for this publication was to use concurrent ambient measurements of ultrafine particles and gaseous precursors to particle formation as evidence to investigate the hypothesis that different gaseous species are more important than others for forming particles at this measurement site. Another important feature of this publication included the investigation of nitrate CIMS as a novel method for measuring ammonia, amines and iodine-containing compounds. Analysis of the variability of mechanisms controlling new particle formation in different areas where the chemical composition of the lower atmosphere is influenced heavily by unique emission sources, such as feedlots, is a pursuit with high scientific merit. The authors attempt to analyze how different sources could be contributing to NPF well by exploiting the capabilities of the nitrate CIMS technique and considering contributions of HOMs to NPF potentially sourced from a local mixed forest. This manuscript is recommended for publication. Included are general comments, technical suggested corrections, and recommendations for improvements of analysis and the method of measurement for future studies.

Comments on analysis and suggestions for improvement:

(1) One of the things that is pointed out in the introduction is that new measurement techniques for amines have appeared over the past few years and it is unclear how spatially variable amines are because it is uncertain how well these techniques actually, quantitatively, measure amines. I think future studies should include a more rigorous approach at an attempt to calibrate this technique to different amines. Attempts at a rigorous calibration of amines for this technique would help to understand, with greater confidence, the spatial variability in concentrations amines around the area of study in this publication, as well as, set a higher standard for measurement of these species by other users of this nitrate CIMS technique. A more rigorous approach to calibration would help advance the understanding of the abundance of atmospheric amines globally, in addition to advancing the understanding of how significant of a role atmospheric amines play in NPF.

We agree with the referee that calibration regarding the amine measurements is essential for extending the knowledge on the spatial and temporal variability of atmospheric amine mixing ratio. We consider an automated on-site calibration procedure, e.g. by using permeation tubes (see Freshour et al., 2014, AMT), as ideal for such a purpose.

(2) Line 154 It would be helpful to add in chemical equations describing the ionization reactions thought to dominate ionization chemistry using the nitrate CIMS technique that are additions to the descriptions on line 154.

Reactions (R1) and (R2) were added to Section 2.2.

(3) Line 170 "while assuming the same ionization efficiency as for sulfuric acid, which has been argued to be a valid assumption" . . . In the supplemental, or appendix?, of Ehn (2014) the authors present the results of quantum computations of binding energies and calculate theoretical collisional rate constants for ELVOCs that are within the measured error for sulfuric acid. Additionally, they compare these calculated rate constants to the measured rate constant based on the sensitivity of the nitrate CIMS measurement to a perfluoroacid. This analysis presented in Ehn (2014), in my opinion, is really good because authentic standards of HOMs are not available. Additionally, because the quantum calculations used structure-activity relationships to derive theoretical dipole moments for some theoretical HOM structures, the results suggesting that clustering of HOMs occurs at the collisional limit only applies to HOMs. It seems one of the primary motivations for the publication of the mass-dependent transmission efficiency study by Heinritzi (2016) was to characterize these effects for HOMs. I believe quantification

of amines using nitrate CIMS through direct calibrations would greatly improve the confidence in the measurement technique, and may even help further support previous attempts at quantifying HOMs with this technique if many directly calibrated compounds appear to undergo ionization at the collisional limit. The importance of direct calibrations is further underscored through the authors suggestion that the nitrate CIMS technique may be particularly less sensitive to ammonia than other compounds they are able to measure.

We agree that the arguments laid out by Ehn et al. (2014) convincingly show that HOMs cluster with nitrate ions at the collisional limit (similar to the deprotonation reaction occurring for sulfuric acid). In this context, we reworded the last part of the sentence in line 171 to:

"… which has been shown to be a valid assumption by Ehn et al. (2014)."

Performing calibration measurements for a variety of compounds measured by nitrate CIMS is one of our main goals for the near future. From our measurements, it seems likely that ammonia is not ionized at the collisional limit. Due to the wide variety of different amines being present in the atmosphere (with different chemical properties), different amines could also be ionized with different sensitivities. In this respect, it should also be noted that sensitivities regarding a certain compound depend also on the settings of the CI-APi-TOF since declustering processes can occur within the instrument. For this reason, especially for the less-stable reaction products calibration of each individual instrument should be performed.

(4) Line 296 An argument is made here that a simpler model can be used to estimate sulfuric acid concentration because the results from this model consistently agree with the results from a more detailed model, which includes RH and CS as variables, within a factor of three. Mikkonen (2011) also points out the minor dependence of the approximated sulfuric acid concentrations on CS. The authors suggest a simplified equation in the conclusions which is only dependent on radiation, the rate constant, and $SO_2$ mixing ratio if particle data is not available. I would argue that if a modeled approximation, and not a measurement, is being used to represent data then it should be treated as rigorously as possible. Including the RH and CS as sinks for $SO_2$, even if they contribute in minor ways, I think is important when forming this approximation, if the data is available. To constrain the approximated data by a factor of three, I think, is a worthwhile action that the authors of the current manuscript took when using the detailed sulfuric acid approximation equation.

We agree with the referee that the parameterization including all parameters ($SO_2$, radiation, RH and CS, see equation (2)) should be used whenever these parameters are available. In order to highlight this importance, the end of Section 3.3 was rewritten as follows:

"This indicates that even the simpler method (equation (3)) can yield relatively accurate results for the conditions of this study. This is probably due to the fact that *RH* and *CS* show only relatively small variations over the duration of the campaign and it would therefore not be absolutely necessary to include these factors in the sulfuric acid calculation. Nevertheless, whenever the data are available we recommend to use the more detailed parameterization (equation (2)) as it treats the sulfuric acid concentration calculation more rigorously."

The discussion section provides some constructive suggestions for improving future measurements! Technical suggestions: **Suggestions for technical corrections ARE IN ALL CAPS

(5) Line 25 "WE DEMONSTRATE HERE that the nitrate CI-APi-TOF is suitable for sensitive measurements. . ."

The sentence in line 25/26 has been modified as suggested:

"We demonstrate here that the nitrate CI-APi-TOF is suitable for sensitive measurements of sulfuric acid, amines, a nitrosamine, ammonia, iodic acid and HOM."

(6) Line 114 ". . .the site is not too far away from the University of Frankfurt, which allowED US to visit the station for instrument maintenance" . . . That sentence either needs to be modified or changed by the above suggestion because right now, grammatically, it doesn't make any sense.

The sentence (lines 112 to 117) has been modified as suggested:

"The site was chosen for several reasons: (i) three larger dairy farms are close by, which should possibly enable us to study the effect of amines on new particle formation, (ii) it can be regarded as typical for a rural or agricultural area in central Europe, (iii) the site is not too far away from the University of Frankfurt, which allowed to visit the station for instrument maintenance on a daily basis and (iv) since we could measure right next to a meteorological station infrastructure and meteorological data from the DWD could be used."

(7) Line 121 "As mentioned in the introduction, livestock ARE known to emit a variety of amines as well as ammonia. . ."

The sentence (lines 121 to 123) has been modified as suggested:

"As mentioned in the introduction livestock are known to emit a variety of amines as well as ammonia (Schade and Crutzen, 1995; Sintermann et al. 2014) both of which should have an influence on new particle formation and growth (Almeida et al., 2013; Lehtipalo et al., 2016)."

(8) Line 303 . . . Please make an explicit statement of the reasons that it was useful to estimate the OH radical concentration as the introduction to this section. This would help to disambiguate the meaning of "further data evaluation", but also help guide, and preview to, the reader to the logic of what the authors saw as the ultimate purpose of doing analyses using estimated OH radical concentrations. This should be a quick and easy modification.

An introductory sentence was added to the beginning of Section 3.4 to explain why it was useful for this study to estimate an OH concentration:

"In this study the hydroxyl radical concentration is required to derive an estimated concentration of the iodine species OIO (Section 3.5) and for a comparison of conditions during nucleation and no nucleation days (Section 4)."

(9) Line 309 . . . Please explicitly define the quantity $[OH]_{day}$. Adding the "day" notation to this quantity makes the exact meaning confusing. I believe this quantity just refers to the steady-state approximated concentration of OH radicals at some time that it is calculated. A similar quantity, $OH_{day}$, is used in the PAM/OFR literature to refer to an estimate of the day-averaged expected OH exposure.

We thank the referee for pointing out a missing definition of the quantity $[OH]_{day}$. By the addition of the subscript "day" we meant to highlight that quite accurate OH concentrations can probably only be calculated during day-time when the photolytic production of $H_2SO_4$ dominates over the production channel via sCI (equation (4)). However, we came to the conclusion that the calculation of [OH] with equation (4) should be possible for the whole day and we have added our arguments to Section 2.4. In addition, the subscript "day" is not being used anymore.

(10) Line 339 . . . "by tentatively adopting the same calibration constant for iodic acid AS sulfuric acid"

The change has been made:

"… by tentatively adopting the same calibration constant for iodic acid as for sulfuric acid."

(11) Line 367 . . . I believe the idea reported by the authors that "iodic acid (and, if present, probably also its clusters) can be detected with high sensitivity due to the high negative mass defect of the iodine atom" is inaccurate. Sensitivity is formally defined as the change in instrument response with respect to a change in some amount of analyte. The sensitivity of the CIMS measurement to any compound is a function of the ionization chemistry and ion transmission through the instrument. What the authors are describing here is the one of the reasons why iodide ionization is a valued as an ionization method in CIMS measurements. The high negative mass defect of iodine produces a characteristic pattern of appearance in the mass spectrum which is useful when trying to unambiguously identify peaks when performing high resolution peak fitting, but this feature doesn't have any reflection on the sensitivity of this particular CIMS measurement to iodic acid. If the authors are measuring the concentrations of iodic acid that they estimate to be measuring then nitrate CIMS is probably particularly sensitive to some iodine compounds, but the logic reported on line 367, I believe, is misleading.

We agree that the arguments at the end of Section 3.5 are somewhat misleading. To clarify these we have reformulated the relevant section (see below). The revised section should now explain better that iodic acid can be unambiguously identified due to its strong negative mass defect. However, the high mass defect also contributes to a low detection limit because there are essentially no other (isobaric) signals that can partly overlap with the one from iodic acid.

"Regarding the sensitivity of the CI-APi-TOF it can be said that iodic acid (and, if present, probably also its clusters) can be detected with high sensitivity. One aspect that helps in unambiguously identifying iodic acid is the high negative mass defect of the iodine atom ($\Delta m \approx$ -0.1 Th). Furthermore, this also contributes to the low detection limit for this compound because generally there will not be any overlapping signals from other substances having the same integer mass (mass resolving power of the instrument is ~4000 Th/Th, i.e. at $m/z$ 175 the peak width at half maximum is ~0.04 Th). The method introduced here therefore allows high-sensitivity measurement of [$HIO_3$] as well as the estimation of [$OIO$] with the help of equation (5) in future studies. The lowest detectable concentrations should be around $3\times10^4$ molecule cm$^{-3}$, or better, for [$HIO_3$] and $5\times10^5$ molecule cm$^{-3}$ for [$OIO$] when assuming the same calibration constant for $HIO_3$ as for $H_2SO_4$ and considering the lowest iodine signal from Fig. 3."

(12) Line 375 . . . "the same calibration constant for iodic acid AS sulfuric acid"

Has been changed (see comment above).

**Additional changes made:**

- Fig. 3, Fig. 5, and Fig. 8: x-axis has been changed to show actual times and not seconds.
- Section 3.6: the explanation for the formation mechanism of NDMA was revised because it is not via a gas-phase reaction between DMA and HONO; instead DMA reacts with OH and NO; the references Pitts et al. (1978), Glasson (1979) and Grosjean (1991) were replaced by the reference to Nielsen, Herrmann and Weller 2012)

**References**

Nielsen, C. J., Herrmann, H., and Weller, C.: Atmospheric chemistry and environmental impact of the use of amines in carbon capture and storage (CCS), Chem. Soc. Rev., 41, 6684–6704, doi: 10.1039/c2cs35059a, 2012.

---

## Author Comment (AC2) · 26 Sep 2016

We thank the referee for the constructive comments, which are added in full below (in black font). Our replies are given in blue font directly after the comments, text that has been added to the manuscript is shown in red font.

Referee #1:

Measurements of sulfuric acid, amines, ammonia, and VOC oxidation products are reported in connection with observations of atmospheric new particle formation (NPF) at a rural site in Germany. Focus of the manuscript is on showing that the nitrate CI-APi-TOF instrument is capable of measuring in ambient conditions ammonia, amines and oxidation products of organic compounds. These have been identified in recent laboratory studies to enhance sulfuric acid–water nucleation rates to atmospheric levels, therefore detecting them in atmospheric measurements is highly relevant for the current nucleation research. The measurements of sulfuric acid are further evaluated by comparing to steady-state proxy concentration. Reporting the proxy coefficients for this environment provides valuable information, since ambient measurements of sulfuric acid are rare, and the proxies have been widely used in different environments.

The CI-APi-TOF data is used to make comparisons between days with and without occurrence of NPF, in order to find out which precursor species affect NPF at this site. No definite participation of ammonia, amines or HOMs to nucleation at this site could be made, but the possible reasons for this are adequately discussed in the manuscript. Also comparisons to chamber measurements from the CLOUD experiment are made. The manuscript is well suited for publication in Atmos. Chem. Phys. I have listed some minor comments and correction/clarification suggestions below (in addition to those made by the anonymous referee 2).

Minor and technical comments:

(1) Line 35: "… a larger fraction …" should be "… a large fraction …"

Done.

(2) Line 78: Please add the abbreviation HOM here, as it is used later in the text.

Done.

(3) Lines 208–210: What are the detection limits for the $SO_2$, $O_3$ and $NO_x$ monitors? In section 3.2, the lowest $SO_2$ concentrations of 0.05 ppb sound quite low for a standard $SO_2$ monitor.

The detection limit for $SO_2$ is reported as 50 pptv (= 0.05 ppbv, for an integration time of 5 minutes) by the company. For the same instrument an even lower detection limit has been reported by Berresheim et al. (2014, ACP) for an integration time of 1 h. The information about the detection limits of the instruments have been added to the beginning of Section 2.4:

"The detection limits of the gas monitors are 0.05 ppbv for the $SO_2$ monitor (for a 5 minute integration time), approximately 0.5 ppbv for the $NO_x$ monitor and 0.5 to 1 ppbv for the $O_3$ monitor."

(4) Line 202: Does this mean the reaction rate constants for the proton transfer reaction in the PTR-MS are similar for different monoterpenes, and therefore are detected with similar efficiency as alpha-pinene?

Yes, the different monoterpenes are detected with similar efficiency by the PTR-MS. Since they all have the same mass (they are mainly detected at a mass to charge ratio of 137 Th, i.e. $C_{10}H_{17}^+$) the PTR-MS cannot distinguish between the different monoterpenes and therefore only the total monoterpene mixing

ratio can be reported. However, α-pinene generally accounts for the largest fraction among all the different monoterpenes. We feel that this is sufficiently explained in Section 2.3 and therefore did not make any adjustment to the text.

(5) Lines 242–244: Please check whether it was 6 or 7 NPF days during the campaign. In Section 3.9 (line 590) it is said 7 events and also Fig 9 shows seven J values.

NPF rates are reported for 6 campaign days. However on one campaign day there were 2 clear particle formation events; therefore 7 rates are reported. We have added this information to the text to avoid confusion. The following information was added to the beginning of Section 3.9:

"It should be noted that clear NPF was observed only on 6 days, however, for one day two NPF rates were derived, which results in a total of 7 NPF rates."

(6) Line 600: Why is the condensational growth out from the 2.5–10 nm size range not considered in Equation 7? That is an additional loss term for particles in this size range, so the right hand side of Eq 7 should have an additional term GR/(7.5 nm)*N (see Kulmala et al. (Nature Protocols 7, 1651–1667, 2012), Equation 9).

The referee is correct. The growth term was accidentally neglected. Including this term does not change the formation rates significantly (on the order of a few tens of percent, only for two events by a factor of ~2). However, the term should of course be included and it was considered in the revised version of the manuscript. Regarding the interpretation of the NPF rates this modification does not lead to any different conclusion

In the context of this comment the reference to Kulmala et al. (2012, Nature Prot.) was added.

(7) Caption text of Figure 1: Please add a mention that the arrows in the bottom panel indicate NPF days. Also check whether there should be six or seven days marked as NPF days (Fig. 9 shows J rates for seven days).

Done (see also reply to comment (5)).

**Additional changes made:**

- Fig. 3, Fig. 5, and Fig. 8: x-axis has been changed to show actual times and not seconds.
- Section 3.6: the explanation for the formation mechanism of NDMA was revised because it is not via a gas-phase reaction between DMA and HONO; instead DMA reacts with OH and NO; the references Pitts et al. (1978), Glasson (1979) and Grosjean (1991) were replaced by the reference to Nielsen, Herrmann and Weller 2012)

**References**

Berresheim, H., Adam, M., Monahan, C., O'Dowd, C., Plane, J. M. C., Bohn, B., and Rohrer, F.: Missing $SO_2$ oxidant in the coastal atmosphere? – observations from high-resolution measurements of OH and atmospheric sulfur compounds, *Atmos. Chem. Phys.*, 14, 12209–12223, doi:10.5194/acp-14-12209-2014, 2014.

Kulmala, M., Petäjä, T., Nieminen, T,, Sipilä, M., Manninen, H. E., Lehtipalo, K., Dal Maso, M., Aalto, P. P., Junninen, H., Paasonen, P., Riipinen, I., Lehtinen, K. E. J., Laaksonen, A., and Kerminen, V.-M.: Measurement of the nucleation of atmospheric aerosol particles, *Nature Prot.*, 7, 1651–1667, doi: 10.1038/nprot.2012.091, 2012.